# Long-term risk of death after tuberculosis diagnosis and treatment

Thiago Cerqueira-Silva [1,2] ✉, Viviane Sampaio Boaventura [2,3], Enny S. Paixão [1,4], Mauro Sanchez[4,5], Clémence Leyrat[1], Otavio Ranzani [6,7], Mauricio L. Barreto[4] & Julia M. Pescarini[1,4]

Tuberculosis (TB) remains a major societal burden, yet data on long-term mortality following TB diagnosis and treatment are limited. We conducted a nationwide Brazilian cohort study using linked data (2004-2018) to quantify long-term mortality (up to 14 years) following TB. We matched: (i) individuals diagnosed with TB or (ii) individuals who had completed TB treatment to TB-free individuals. We used competing risk methods to analyze natural causes (that is, defined as deaths excluding TB, HIV and external causes) and cause-specific mortality. In the diagnosed cohort (185,921 pairs), the risk of 14-year natural cause mortality was significantly higher (risk ratio (RR) = 2.16, 95% confidence interval = 1.96-2.37); RRs were significantly elevated for deaths due to cancer, cardiovascular, endocrine, respiratory and external causes. The treated cohort (111,871 pairs) presented elevated natural cause mortality risk (RR = 1.77,1.55-2.03), with similarly increased RRs across specific causes. We showed that TB survivors, even after treatment, faced a significantly elevated, prolonged risk of death from various causes up to 14 years later. This finding highlights the need for long-term monitoring to reduce the burden of TB.

Tuberculosis (TB) continues to be one of the deadliest infectious diseases worldwide. In 2023, approximately 10.8 million individuals developed TB, and 1.25 million died from this disease[1].

These alarming figures highlight the critical importance of the World Health Organization's (WHO) End TB Strategy, which aims to reduce TB incidence by 90% and TB mortality by 95% by 2035, relative to 2015 levels[2]. However, progress has been insufficient: by 2023, the global TB incidence has declined by only approximately 8% and TB mortality has declined by approximately 23%[1]. Addressing this gap requires a comprehensive approach that extends beyond treating active TB, considering its long-term impact on health systems through increased demand for chronic disease services and its role in perpetuating health inequities.

Although TB treatment is effective in curing the disease and substantially decreases mortality during the active phase[1], emerging evidence suggests that individuals who survive TB remain at an elevated risk of death long after completing treatment[3–5]. This ongoing risk may stem from factors such as lasting lung damage, chronic inflammation, coexisting health conditions and poor social circumstances[2,5,6]. Despite the magnitude of this issue, the long-term burden of mortality following TB remains mostly overlooked in public health strategies; notably, the WHO guidelines contain no recommendations for addressing post-TB conditions[5].

Studies assessing post-TB mortality have relied exclusively on relative risk measures, without estimating absolute risks, thereby limiting the ability to quantify the excess mortality attributable to TB[6–9]. In addition, most studies on mortality post-TB to date have compared all-cause mortality to that of the general population, adjusting only for age and sex, but without properly controlling for socioeconomic variables associated with the risk of TB and death[6,7]. To overcome these

[1]Faculty of Epidemiology and Population Health, London School of Hygiene & Tropical Medicine, London, United Kingdom. [2]Laboratório de Medicina e Saúde Pública de Precisão, Fundação Oswaldo Cruz, Salvador, Brazil. [3]Faculdade de Medicina da Bahia, Universidade Federal da Bahia, Salvador, Brazil. [4]Centro de Integração de Dados e Conhecimento para a Saúde (CIDACS), Fundação Oswaldo Cruz, Salvador, Bahia, Brazil. [5]Faculty of Health Sciences, University of Brasília, Brasília, Brazil. [6]DataHealth Lab, Institut de Recerca Sant Pau (IR SANT PAU), Barcelona, Spain. [7]Pulmonary Division, Faculty of Medicine, Heart Institute, Hospital das Clínicas da Faculdade de Medicina da Universidade de São Paulo, São Paulo, Brazil. ✉e-mail: thiago.silva@lshtm.ac.uk

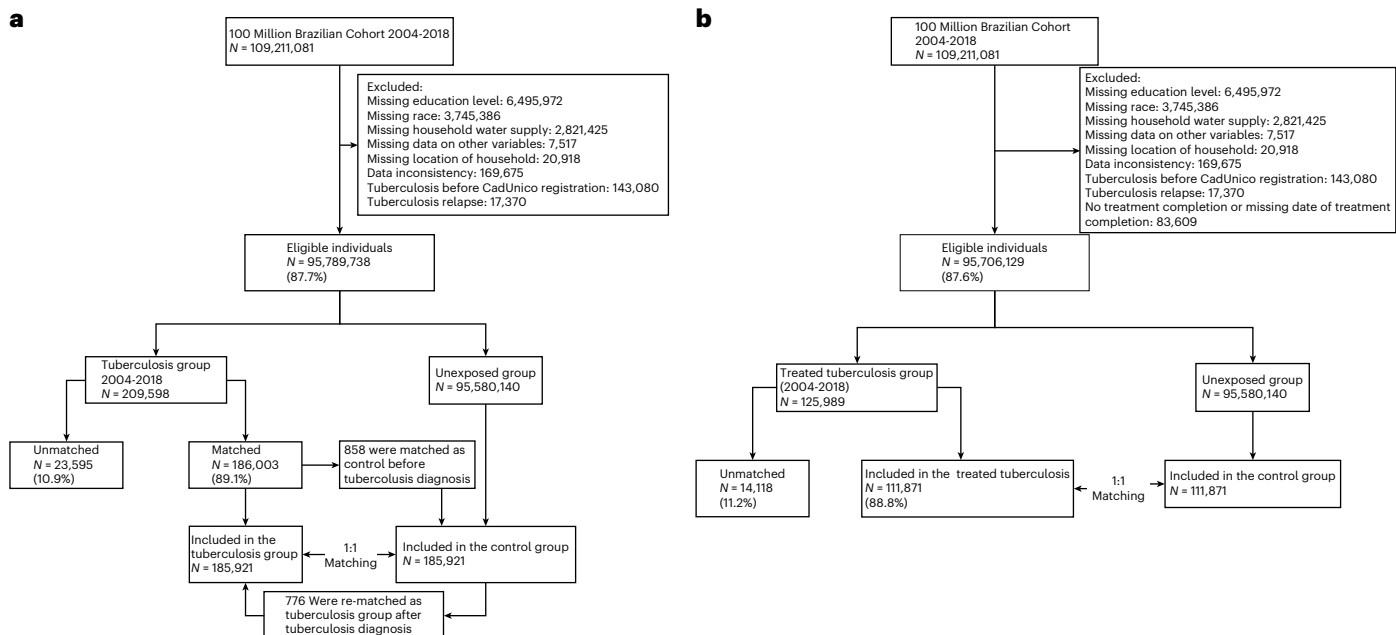

**Fig. 1 | Study flowchart. a,b,** Selection process for the matched cohort for diagnosed TB cases (**a**) and matched cohort for treated TB cases (**b**).

limitations and provide a clearer understanding of the lasting impact of TB, a comprehensive evaluation of adverse outcomes post-TB with greater control of socioeconomic characteristics is needed. Closing this knowledge gap is critical for improving the understanding of post-TB health trajectories and informing preventive and long-term care strategies for affected individuals.

Leveraging nationwide administrative data from the 100 Million Brazilian Cohort, which represents the poorest half of the population of Brazil, this study investigated the risk of death following TB diagnosis and TB treatment completion. Specifically, we (i) compared the risk of death between TB cases after diagnosis and after treatment completion and control participants over time, in terms of natural deaths (defined as deaths excluding TB, HIV, and external causes), all-cause mortality and cause-specific of death (cancer, cardiovascular, etc.), and (ii) assessed whether there was a difference in the risk of death by sex, age, race, and TB classification.

## Results

Among the 95,789,738 eligible people who entered the 100 Million Brazilian Cohort between 2004 and 2018, 209,598 had TB. A total of 185,921 (88.4%) diagnosed TB exposed were matched with an unexposed individual. Among the 125,989 treated TB exposed, 111,871 (88.8%) were matched with an unexposed individual (Fig. 1, Table 1 and Supplementary Table 2).

The unmatched TB-exposed individuals (23,667 (11.6%) of cases diagnosed with TB, and 14,118 (11.4%) of those treated for TB) were older, had a higher proportion of minorities (Asian/Indigenous), resided more in rural areas and had worse socioeconomic conditions (higher proportion of overcrowded houses, with worse access to water) (Supplementary Table 3).

The median follow-up period in the diagnosed cohort was 4.5 years (interquartile range (IQR) 1.9 to 7.7) for the unexposed group and 3.9 years (IQR 1.5 to 7.2) for the diagnosed group. For the treated cohort, the median follow-up was 4.0 years (IQR 1.8 to 7.2) for the unexposed group and 3.8 years (IQR 1.7 to 7.0) for the treated group. The median time to treatment completion was 6.6 months (IQR, 6.2 to 7.7).

In the diagnosed cohort, 29,226 deaths occurred during follow-up, 23,900 (12.6%) in the TB-diagnosed group and 5326 (2.2%) in the unexposed group (Supplementary Figs. 1 and 2). In the treated cohort, 10,631

deaths occurred, with 7,677 (6.6%) in the TB-treated group and 2,954 (2.3%) in the unexposed group (Supplementary Figs. 3 and 4)

### Natural deaths (excluding external, HIV and TB causes)

In the diagnosed cohort, the risk of death within 30 days of diagnosis was markedly higher among TB-exposed participants compared to unexposed participants (risk ratio (RR): 28.18; 95% confidence interval (CI), 21.68 to 37.04), decreasing to 6.68 (6.20 to 7.26) at the end of the first year and 2.16 (1.96 to 2.37) at the end of 14 years (Table 2 and Supplementary Table 4). The yearly estimates (incidence rate ratios (IRRs)) showed values close to 2.4 (ranging from 1.99 to 3.22) between 2 and 10 years. (Fig. 2 and Supplementary Table 4)

In the treated cohort, the risk of death among people treated for TB compared to unexposed participants was more stable over time, with an RR of 2.72 (1.94 to 3.92) within 30 days of treatment completion, 3.33 at 90 days (2.69 to 4.27) and of 1.77 (1.55 to 2.03) at the end of the 14 years of follow-up (Table 2). Similarly to the diagnosed cohort, the yearly IRRs were close to 2.2 (ranging from 2.03 to 2.62) between 2 and 10 years after treatment completion (Fig. 2 and Supplementary Table 5).

### All-cause mortality

In the diagnosed cohort, the all-cause mortality RR comparing the TB group with the unexposed group was markedly higher in the first month after diagnosis (RR: 58.10; 47.05 to 73.04). The RR decreased to 12.66 in 1 year and 2.89 (2.69 to 3.11) in 14 years. At 14 years, the risk difference (RD) per 100,000 persons was 15,167.8 (14,343.2 to 16,015.5) (Supplementary Table 4 and Supplementary Fig. 1).

For the treated cohort, the RR in the first month after treatment completion was 3.22 (2.41 to 4.31), decreasing to 2.01 (1.81 to 2.28) at 14 years, with a RD of 8,206.6 (7,131.3 to 9,453.2) (Supplementary Table 5 and Supplementary Fig. 3).

### Cause-specific mortality

The cause-specific analysis showed a similar pattern to that of the analysis of natural death. In the diagnosed cohort, the highest RRs were observed within 30 days of the diagnosis, which decreased over time (Supplementary Tables 4 and 7 and Supplementary Fig. 2). The cumulative incidence of cardiovascular, cancer and respiratory

**Table 1 | Baseline characteristics of diagnosed and treated TB cases**

| Characteristic | Diagnosed TB | | Treated TB | |
|---|---|---|---|---|
| | Unexposed *n*=185,921 | Exposed *n*=185,921 | Unexposed *n*=111,871 | Exposed *n*=111,871 |
| **Age (y) at index date, median (IQR)** | 33 (24-47) | 33 (24-47) | 33 (25-47) | 33 (25-47) |
| **Age group (y) at index date** | | | | |
| <18 | 12,520 (6.7) | 12,520 (6.7) | 6,963 (6.2) | 6,963 (6.2) |
| 18-59 | 154,081 (82.9) | 154,081 (82.9) | 92,998 (83.1) | 92,998 (83.1) |
| ≥60 | 19,320 (10.4) | 19,320 (10.4) | 11,910 (10.6) | 11,910 (10.6) |
| **Sex** | | | | |
| Female | 68,470 (36.8) | 68,470 (36.8) | 42,829 (38.3) | 42,829 (38.3) |
| Male | 117,451 (63.2) | 117,451 (63.2) | 69,042 (61.7) | 69,042 (61.7) |
| **Race/Ethnicity** | | | | |
| White | 49,599 (26.7) | 49,599 (26.7) | 28,695 (25.7) | 28,695 (25.7) |
| Black | 21,183 (11.4) | 21,183 (11.4) | 12,116 (10.8) | 12,116 (10.8) |
| Mixed | 113,311 (60.9) | 113,311 (60.9) | 69,769 (62.4) | 69,769 (62.4) |
| Asian | 264 (0.1) | 264 (0.1) | 156 (0.1) | 156 (0.1) |
| Indigenous | 1,564 (0.8) | 1,564 (0.8) | 1,135 (1.0) | 1,135 (1.0) |
| **Education level** | | | | |
| No school | 24,436 (13.1) | 24,436 (13.1) | 14,806 (13.2) | 14,806 (13.2) |
| Nursery | 1,503 (0.8) | 1,503 (0.8) | 951 (0.9) | 951 (0.9) |
| Infant school | 1,720 (0.9) | 1,720 (0.9) | 1,033 (0.9) | 1,033 (0.9) |
| Elementary school | 70,454 (37.9) | 70,454 (37.9) | 41,865 (37.4) | 41,865 (37.4) |
| Middle school | 65,170 (35.1) | 65,170 (35.1) | 38,951 (34.8) | 38,951 (34.8) |
| High school | 21,962 (11.8) | 21,962 (11.8) | 13,833 (12.4) | 13,833 (12.4) |
| Higher education | 676 (0.4) | 676 (0.4) | 432 (0.4) | 432 (0.4) |
| **Overcrowded**[a] | 27,968 (15.0) | 27,968 (15.0) | 16,240 (14.5) | 16,240 (14.5) |
| **Location of household** | | | | |
| Urban | 164,096 (88.3) | 164,096 (88.3) | 97,496 (87.2) | 97,496 (87.2) |
| Rural | 21,825 (11.7) | 21,825 (11.7) | 14,375 (12.8) | 14,375 (12.8) |
| **Diabetes** | - | 9,768 (5.3) | - | 5,908 (5.3) |
| **HIV** | - | 15,286 (8.2) | - | 6,031 (5.4) |
| **TB classification** | | | | |
| Pulmonary | | 157,540 (84.7) | | 96,328 (86.1) |
| Extrapulmonary | | 23,134 (12.4) | | 13,110 (11.7) |
| Both | | 5,178 (2.8) | | 2,433 (2.2) |
| Missing | | 69 (<0.1) | | 0 (0) |
| **Laboratory diagnosis** | | 127,838 (68.8) | | 78,239 (69.9) |
| **Radiographic or laboratory diagnosis** | | 174,484 (93.8) | | 105,915 (94.7) |

Sputum culture, histopathology, bacilloscopic or molecular tests were considered as laboratory diagnosis. IQR, interquartile range. [a]Defined as a household density (residents per room) greater than 2.

deaths were similar in the TB-exposed group, whereas the cardiovascular deaths had the highest incidence in the unexposed participants (Supplementary Fig. 2).

In the treated cohort, the highest RR for cancer and respiratory deaths were seen within 90 days of treatment completion (Supplementary Tables 5 and 8 and Supplementary Fig. 4). The cumulative incidence of cardiovascular deaths were slightly higher than cancer and respiratory deaths (Supplementary Fig. 4).

The treated cohort showed lower RRs for all cause-specific deaths, especially for cancer and respiratory diseases; for example, the 1-year RRs for respiratory deaths were 12.72 (10.21 to 16.93) in the diagnosed cohort and 6.15 (4.71 to 8.78) in the treated cohort.

Notably, within the specific cancer types, we found an increased risk for cancer of the digestive organs in both cohorts; the RRs at 14 years were 1.97 (1.55 to 2.57) in the diagnosed cohort and 1.98 (1.40 to 2.89) in the treated cohort (Supplementary Tables 7 and 8).

One exception that presented similar values across both cohorts was death from external causes. In the diagnosed cohort, the RRs ranged from 1.72 to 2.47, and in the treated cohort, RRs ranged from 1.74 to 2.10 (Fig. 3 and Supplementary Tables 4 and 5). In the analysis by ICD-10 blocks of external causes, deaths from assault (homicides) presented higher RRs than deaths from accidents in both cohorts (Supplementary Tables 7 and 8).

## Subgroup analyses
The subgroup analysis for the risk of death due to natural causes exhibited the same pattern as the main analysis, with the RR decreasing over time. In the diagnosed cohort, females had slightly higher RRs

**Table 2 | Estimated risk of death from natural causes (excluding external causes, HIV and TB deaths) of diagnosed and treated TB-exposed participants compared to unexposed participants**

| Timeª | Cumulative number of events | | Risk per 100 000 people (95% CI) | | Risk difference per 100,000 people (95% CI) | RR (95% CI) | E-value (CI) |
|---|---|---|---|---|---|---|---|
| | Unexposed | TB | Control group | TB group | | | |
| **Diagnosed cohort** | | | | | | | |
| 30 days | 54 | 1,524 | 29.2 (22.2-37.3) | 822.4 (782.8-860.9) | 793.2 (753.7-832.9) | 28.18 (21.68-37.04) | 55.85 (42.85) |
| 1 year | 673 | 4,636 | 389.2 (360.5-418.5) | 2,600.9 (2,526.9-2667.0) | 2,211.7 (2,139.2-2,286.0) | 6.68 (6.20-7.26) | 12.85 (11.88) |
| 5 years | 2746 | 9,518 | 2,116.3 (2,033.0-2,197.4) | 6,562.7 (6,433.3-6699.4) | 4,446.4 (4,277.5-4,602.6) | 3.10 (2.97-3.24) | 5.65 (5.39) |
| 10 years | 3892 | 11,847 | 4,327.3 (4,166.6-4,484.2) | 10,945.4 (10,719.2-11,176.8) | 6,618.1 (6,344.0-6,888.7) | 2.53 (2.43-2.64) | 4.50 (4.29) |
| 14 years | 4060 | 12,151 | 6,451.7 (5,886.5-6,975.2) | 13,947.2 (13,413.3-14,553.7) | 7,495.5 (6,652.7-8,257.8) | 2.16 (1.96-2.37) | 3.75 (3.33) |
| **Treated cohort** | | | | | | | |
| 30 days | 39 | 106 | 35.1 (24.3-46.8) | 95.3 (78.2-112.3) | 60.2 (39.1-80.0) | 2.72 ( 1.94-3.92) | 4.88 (3.29) |
| 1 year | 380 | 1,206 | 364.7 (327.1-400.4) | 1,158.2 (1,090.1-1,225.0) | 793.5 (719.7-870.4) | 3.18 (2.83-3.55) | 5.80 (5.11) |
| 5 years | 1578 | 3,950 | 2,111.8 (2,005.8-2,213.4) | 5,093.1 (4,943.8-5,257.1) | 2,981.3 (2,800.1-3,177.8) | 2.41 (2.29-2.55) | 4.26 (4.00) |
| 10 years | 2181 | 5,215 | 4,283.7 (4,080.3-4,510.5) | 9,607.3 (9,290.8-9,930.4) | 5323.6 (4,961.2-5,662.3) | 2.24 (2.12-2.37) | 3.91 (3.65) |
| 14 years | 2258 | 5,336 | 6,655.3 (5,826.3-7,501.2) | 11,756.4 (11,220.3-12,389.8) | 5,101.0 (4,105.4-6,128.4) | 1.77 (1.55-2.03) | 2.93 (2.48) |

ªTime since TB diagnosis or end of treatment.

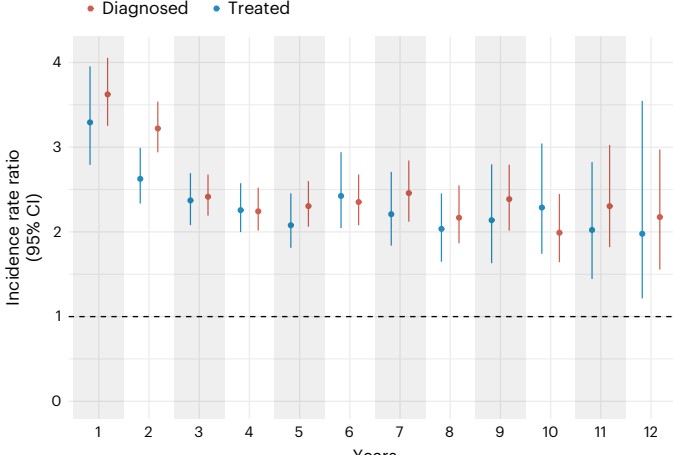

**Fig. 2 | Yearly IRRs for deaths due to natural causes.** IRRs for deaths due to natural causes (excluding HIV, TB and external causes) of diagnosed (red) and treated (blue) TB-exposed participants compared to unexposed participants. Error bars represent 95% CIs. The number of individuals in each group by year of follow-up is displayed in Supplementary Figs. 1 and 3.

than males throughout the entire follow-up period. Participants aged 18-59 years had a higher RR than those aged 60 years or older (RR at 14 years: 2.71 (2.35 to 3.14) and 1.43 (1.28 to 1.60), respectively). However, on the RD scale, participants aged 60 years or older had nearly three times the risk of death compared to those aged 18-59 years. The RD at 14 years for 18-59 years was 7,047.4 (6,165.6 to 7,858.2); and for ≥ 60 years it was 14,900.6 (10,525.5 to 19,344.6). Participants with pulmonary TB exhibited lower RR and RD than those with extrapulmonary or both extrapulmonary/pulmonary TB (Extended Data Fig. 1 and Supplementary Table 9).

In the treated cohort, males and females had similar RR. The age group pattern was maintained, with the 18-59 age group showing a higher RR and lower RD than those ≥60 years. In this cohort, participants with pulmonary TB exhibited similar values of RR and RD than those with extrapulmonary or both extrapulmonary/pulmonary TB (Fig. 4 and Supplementary Tables 11).

In both cohorts, we did not observe any substantial differences in the temporal patterns or baseline risk magnitude by race/ethnicity. Participants with diabetes mellitus (DM) had a similar RR to those with HIV in the diagnosed cohort and a higher RR in the treated cohort. Notably, the absolute excess mortality (RD) in patients with TB and DM was substantially larger, exceeding double the RD observed in patients with TB and HIV (Fig. 4, Extended Data Fig. 1 and Supplementary Tables 10 and 11).

We also evaluated the cause-specific mortality by sex. Both cohorts (diagnosed and treated) showed similar RR for cardiovascular, respiratory and endocrine causes for males and females, whereas the RR for cancer deaths was slightly higher in males than in females (Supplementary Tables 12-15).

**Household contacts**

A total of 448,825 household contacts were matched to unexposed participants. The risk of death among the TB household contacts was elevated when compared to their unexposed counterparts. Within 5 years, the RR for all-cause mortality was 1.19 (RR 1.14 to 1.23), for natural deaths was 1.11 (1.06 to 1.16), and for external deaths it was 1.29 (1.20 to 1.39). At 14 years, the RR for all-cause mortality was 1.09 (1.00 to 1.19), for natural deaths 1.04 (0.93 to 1.15) and for external causes of death 1.16 (1.00 to 1.34) (Supplementary Tables 6 and 9).

In the second analysis, a total of 12,948 (6.2%) TB cases were matched to household contacts. The median age of the pairs was 14 years (IQR 11 to 18), and 8,980 pairs (69.4%) were male. The direct comparison between TB cases and household contacts for natural

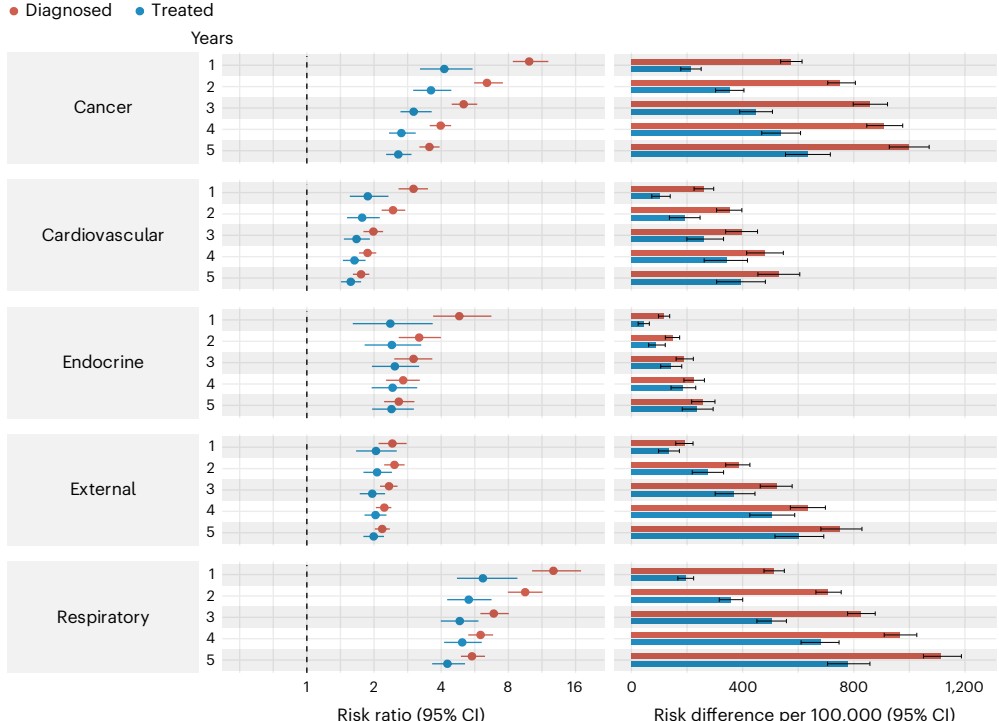

**Fig. 3 | Cause-specific risk of death of patients with diagnosed and treated TB.** Estimated cause-specific risk of death of diagnosed and treated TB-exposed participants compared to the unexposed group in the first 5 years. Causes of death defined by ICD-10 codes: (i) Cardiovascular causes (Chapter IX), (ii) Endocrine causes (Chapter IV); (iii) Respiratory System (Chapter X); (iv) Cancer (Chapter II); (v) External Deaths. Error bars represent 95% CIs. The x axis for the RR is in the log scale. The number of individuals in each group by year of follow-up is displayed in Supplementary Figs. 1 and 3.

deaths yielded results similar to those of the main analysis, with an RR of 11.15 (6.49 to 23.10), 3.97 (2.94 to 5.70) and 2.52 (1.83 to 3.66) at 1, 5 and 10 years, respectively. For external causes of death, the RR at 1, 5 and 10 years was 2.29 (1.48 to 4.09), 1.66 (1.29 to 2.12) and 1.40 (1.06 to 1.84), respectively (Supplementary Table 16).

## Discussion

In this large, population-based cohort study conducted in Brazil, we found a significantly elevated risk of mortality among individuals diagnosed and treated for TB compared with TB-free participants with similar socioeconomic characteristics. Over 14 years of follow-up, patients diagnosed with TB had 15,168 more deaths per 100,000 persons compared with TB-free participants. After treatment, we observed a decrease in the RD, but it remained substantial at 8,206 more deaths per 100,000 persons. Our findings reveal that individuals successfully treated for TB still experience excess mortality across multiple organ systems and causes, highlighting the lasting impact of TB on overall health.

Previous studies have also evaluated mortality following TB[6–12]. However, our study provides finer control over socioeconomic characteristics, particularly for having detailed information for the TB-free participants. Controlling for socioeconomic variables is extremely important when evaluating the residual burden of TB, as the heightened vulnerability of individuals who develop TB also contributes to the increased risk of mortality compared with the general population, partly due to poverty[13]. We have also conducted a comprehensive evaluation of mortality risks using a robust methodology, which includes exact matching of multiple characteristics within a competing risks framework for individuals with a TB diagnosis and those who have completed TB treatment. Second, we have presented both absolute and relative measures for risks and the rate ratios by year of follow-up in all comparisons. These measures complement one another, offering a thorough view of the TB burden. Third, we have assessed the risks

after confirmed treatment completion, with clinical or microbiological confirmation of treatment success, rather than defining the post-TB period as 12 months following diagnosis, as in previous studies[6,7]. Relying solely on time after diagnosis can group individuals who abandoned treatment, experienced treatment failure, who are classified as having drug-resistant TB, or who were lost to follow-up with those who completed treatment, likely biasing the results[6]. In our study, half of the patients completed their treatment at 199 days post-diagnosis, with most finishing treatment before 8 months. Finally, we used natural causes deaths as the main outcome and excluded TB and HIV related deaths. The curated outcome definition and use of TB cases with proof of treatment completion allow us to quantify the residual burden of TB with greater accuracy.

Our findings showed that diagnosed and treated TB cases had a higher risk of mortality across a broad range of causes, including respiratory, cardiovascular, endocrine and cancer. Our results complement previous evidence showing that TB is associated with an increased risk of respiratory mortality, mainly due to direct lung damage caused by TB, increasing the risk of recurrent pneumonia and chronic obstructive pulmonary disease, bronchiectasis and other specific infections, such as aspergillomas[3,14]. Our findings of sustained increased risk of cardiovascular deaths post-TB diagnosis for more than a decade complements previous epidemiological studies assessing this association on the short-term scale[7,15,16].

Additionally, endocrine-related deaths were notably increased, potentially reflecting a bidirectional relationship between diabetes mellitus and TB[17,18]. Diabetes can predispose individuals to TB through impaired immune responses, whereas chronic inflammation from TB can exacerbate insulin resistance and metabolic dysregulation, thereby increasing the risk of diabetes-related mortality[18,19]. Shared inflammatory mechanisms and dysregulated immune responses likely underpin the mutual exacerbation observed between these two conditions[18,20].

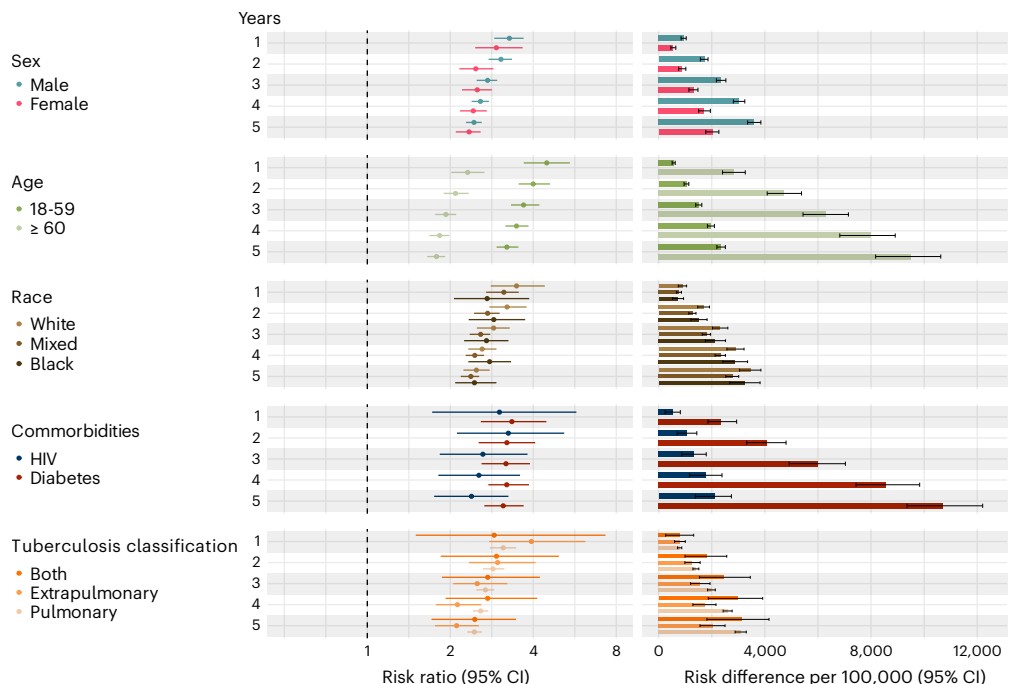

**Fig. 4 | Stratified analysis for the treated TB-exposed group.** Stratified estimated risk of death from natural causes (excluding external causes, HIV and TB deaths) of treated TB-exposed compared to the unexposed group. In TB classification, "both" indicates pulmonary plus extrapulmonary. Error bars represent 95% CIs. The x-axis for the RR is in the log scale. The number of individuals in each group by year of follow-up is displayed in Supplementary Fig. 3.

Our findings showed that TB is associated with an increased risk of deaths from cancer, including cancer in the digestive organs. This finding is consistent with a growing body of evidence; notably, a prior meta-analysis of 11 studies showed an elevated risk of cancer in patients with TB for more than 5 years after diagnosis[21]. Although the mechanisms underlying this association are not fully understood, several hypotheses have been proposed, such as chronic systemic inflammation that can promote carcinogenesis through the promotion of reactive oxygen species and DNA damage[22–24]. The increased risk of deaths from cancer may also be partially explained by a higher prevalence of shared lifestyle risk factors for cancer and TB, such as smoking and alcohol use[8,25–27]. Our work extends this body of evidence by providing detailed, year-by-year risk estimates, highlighting the significant long-term outcomes of TB and supporting the need for continued patient care for TB-related sequelae after successful treatment.

We also observed an increased risk of deaths due to external causes, with similar magnitudes in the diagnosed and treated TB cases. Although we cannot dismiss the possibility that this finding is due to residual confounding, as there is no biological plausibility for this relationship, it may also reflect the social stigma experienced by patients with TB. This stigma can lead to social isolation, limited economic opportunities, and potentially worsening underlying or new-onset mental health issues such as depression or anxiety, which, in turn, could promote riskier behaviors[28–30]. However, there is also the possibility that poor mental health or living in contexts of violence could increase the chances of TB reactivation[31] and further lead to increased external causes-related mortality after TB diagnosis or treatment. However, similarities in the RR between the treated and diagnosed cohorts reinforce the plausibility of these increased risks being due to a societal rather than biological phenomenon. In addition, our direct comparison of TB cases to household contacts of the same sex and similar age also showed elevated risk of deaths from external causes in the TB group. This finding indicates that, even under similar socioeconomic conditions, patients with TB experience an increased risk of death from external causes. Although some of this association can be related to residual confounding, it is unlikely to account for all of it. In this context, increased awareness of TB stigma and

interventions to address it, such as TB support groups, training healthcare workers to provide nonjudgmental care, and community-wide educational campaigns to dispel myths and misinformation about TB can improve the lives of TB survivors[32].

In our study, we observed a slight increase in the risk of death among household TB contacts compared with unexposed individuals, suggesting some degree of residual confounding likely due to heightened social vulnerability in households with a TB case. However, the excess risk of death among patients diagnosed with, and even those treated for, TB far exceeds the risk found among household contacts. This suggests that although there are indeed connections between poverty and TB[13], the increased mortality after TB treatment cannot be attributed solely to poverty, as indicated in previous studies[4].

Our study has several limitations. First, like any observational study, it is susceptible to residual confounding. We also could not account for potential time-varying confounders after the index date, such as loss or decrease in family income. Our attempt to quantify residual confounding involved conducting an additional analysis to evaluate mortality among household contacts compared to people without TB using the same approach as the main analysis. Although our intent was not to estimate the risk of death in contacts itself, contacts share similar socioeconomic and housing conditions to people with TB, and we do not expect a strong causal link between being a TB contact and an increased risk of death. This analysis revealed a slightly elevated risk of death in this group. However, the excess risk of death among patients diagnosed with and even those treated for TB far exceeds the risk found among household contacts.

We also estimated E-values, which quantify how strong unmeasured confounding would need to be to explain away an observed association, to assess the robustness of our findings, the E-value for the RR at 10 years was 4.50 for the diagnosed cohort and 3.91 for the treated cohort, considering the strength of confounders in previous studies, the only confounder with effect measures associations greater than 4.00 was age, which was accounted in our study, indicating that the TB-mortality association found in our study is unlikely to be fully explained by unmeasured confounders[6,8,33].

Additionally, our analysis is restricted to the poorer half of Brazil. Although this may limit generalizability to wealthier populations, it likely enhances control for confounding factors due to socioeconomic characteristics by studying a more homogenous group and focusing on a highly vulnerable population where the impact of TB is often most severe. Nevertheless, even in this large cohort, we were unable to match all participants, with the unmatched group exhibiting worse socioeconomic conditions than the matched sample. Our decision to match increases internal validity, providing adjustment by design without making any assumptions about the exposure or outcome model. However, even with a higher percentage of matched cases (>88%), matching may lead to an underestimation of TB effects, considering that the unmatched group had poorer socioeconomic conditions, and likely a higher baseline mortality risk. The presence of unmatched cases also changes the estimate of the average exposure effect in the exposed to average exposure effect in the matched sample[34].

Second, our results may be subject to reverse causality, particularly for cancer and endocrine-related mortality. Given that active malignancies and diabetes are known risk factors for TB and that we were unable to adjust for pre-exposure comorbidity, it is possible that an undiagnosed underlying condition precipitated the onset of active TB in some individuals[14,19]. In such cases, TB would be a marker of underlying vulnerability or an exacerbating factor, rather than the primary cause of development and death from cancer or endocrine diseases. Although our long-term follow-up and analysis of the treated cohort mitigate this concern for deaths occurring many years after diagnosis, the possibility cannot be fully excluded, especially for mortality observed in the early follow-up period. Third, we only had comorbidity data on HIV and DM for the TB group, which were assessed at the time of TB diagnosis via the SINAN system. This information was absent for the unexposed participants in the CadÚnico database. Consequently, our subgroup analyses estimate the joint effect of TB plus comorbidities, rather than the effect of TB alone. This prevents us from disentangling the independent effect of TB from the effect of these pre-existing comorbidities. Fourth, only 15% of the TB cases had sputum culture positive, which is considered the gold standard for the diagnosis of TB. This proportion reflects the Brazilian guidelines, which prioritizes sputum culture for suspected TB cases that present a negative bacilloscopy, and specific cases such as suspected cases of resistance, retreatment cases, in prison populations[35]. In addition, due to the required infrastructure and time for diagnosis, the use of sputum culture as primary diagnosis for TB remains limited in low-and-middle income countries[36,37]. The inclusion of nonconfirmed TB cases in the TB group likely underestimates the risk in this group.

Fifth, we lack detailed information (type of resistance) about the TB cases that were identified as drug-resistant (*n* = 945; 0.5%) after completing the treatment scheme, considering the very low numbers it is unlikely to significant bias the analysis of diagnosed cases, and it has no effect in the analysis of treated cases, as these TB cases were not included. Sixth, we lack information about latent TB infection (LTBI), which can result in some individuals with LTBI being classified as TB-free unexposed participants, biasing our estimates downwards as previous studies have shown that LTBI slightly increases the risk of death compared to healthy individuals[38]. This also represents that our results reflect excess mortality following a TB disease episode, not the isolated biological effect of TB. Seventh, after 11 years of follow-up, we had a relatively small number of participants at risk, resulting in a limited number of events and imprecise estimates by risk period. This issue also occurred in the subgroup analysis, requiring caution when interpreting the point estimates from those analyses. The subgroup and cause-specific analyses were not adjusted for multiplicity adjustment, as specified in the analysis plan, and should not be used in place of hypothesis testing. Eighth, we relied on International Classification of Diseases, 10th edition (ICD-10th) codes from the mortality information system to classify the underlying cause of death, whereas there is

no comprehensive study evaluating the quality of this codification in Brazil. The quality of the death certificate information assessed by the percentage of garbage codes has improved nationwide, with a reduction of up to 90% of garbage codes between 2000 and 2015 (ref. 39). Lastly, we were unable to censor individuals when they emigrated from Brazil. However, emigration rates for Brazil are low, estimated at 0.8% in 2019 (ref. 40), making it unlikely to substantially bias our estimates.

One fundamental limitation about the management of TB worldwide is the sole focus on diagnosing and curing active disease, overlooking long-term health consequences. For decades, WHO guidelines have appropriately emphasized the diagnosis and bacteriological cure of active disease, considering this a complete return to health, without considering possible post-TB complications. Our findings strongly support the need for long-term clinical follow-up into routine TB care. Integrating post-TB assessments, such as lung function testing, cardiovascular risk screening, and cancer surveillance, into national guidelines for post-TB management is essential. Such measures will increase clinician awareness of post-TB complications, ensure timely management, and direct resources towards truly comprehensive, patient-centered care.

## Online content

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

## Methods

We used data from the 100 Million Brazilian Cohort linked with nationwide death and TB registries. The 100 Million Brazilian Cohort is a dynamic cohort comprising over 130 million individuals from the Unified Registry for Social Programs (CadÚnico). CadÚnico serves as Brazil's primary tool for identifying and registering low-income families (that is, families with monthly per capita income less than half minimum wage (approximately $284 USD in 2025) and enrolling them in eligible social welfare programs. Thus, it captures predominantly individuals from the lower socioeconomic strata of the country. We linked the CadÚnico database to TB disease records from 1 January 2004 to 31 December 2018, registered in the National Notifiable Disease Information System (SINAN) and the Mortality Information System. The Mortality Information System has an estimated completeness of 98% of all deaths occurring in Brazil[41].

In Brazil, TB is a mandatory notification disease, with diagnosis made by rapid molecular test for TB, bacilloscopy, sputum culture, thorax X-ray or clinical case definition[42]. All suspected and confirmed cases of TB should be registered with SINAN via a notification and follow-up form that contains sociodemographic and clinical information about the individuals, including information about being a newly diagnosed TB case or reinfection case and also data about the treatment outcome. This should be filled in by a healthcare professional, usually a nurse or a medical doctor.

The linkage between the two databases was done using CIDACS-RL[43], a linkage tool based on a similarity index between registries in different databases. The linkages were based on five variables (name of the individual, name of the mother, date of birth, sex and municipality of residence). The linkage between TB registries and the CadÚnico had a sensitivity (94.6%, showing the proportion of true TB registries classified as TB) and specificity (93.6%, showing the proportion of non-TB registries classified as non-TB) calculated based on false or true links between the two databases. The linkage between mortality registries and the CadÚnico calculated by year had a sensitivity that ranged between 97.8% and 100.0% and a specificity between 96.6% and 99.9% depending on the year of death, details about this linkage can be found in previous article[44]. Registration in CadÚnico was used to define our baseline study population independent of individuals updating or not the registry after, whilst exposure definition was based on the linkage with TB administrative records and follow-up based on linkage with mortality records. Therefore, from the linked dataset, we built two cohorts: one matching persons with a TB diagnosis to TB-free unexposed participants, and another matching participants who had completed TB treatment to TB-free unexposed participants.

### Exposure and outcomes

The exposure was an individual's first SINAN record of new TB between Jan 1, 2004, and Dec 31, 2018 for the diagnosed cohort and an individual's first SINAN record of treatment completion for TB between Jan 1, 2004, and Dec 31, 2018 for the treated cohort. We defined treatment completion as being classified as "treatment success" in the SINAN. Patients with treatment completion are those who have two negative smear tests according to national guidelines or who do not have evidence of treatment failure based on clinical or radiological criteria[42]. Unexposed individuals were those TB-free and alive during the study period.

To test our hypothesis that people previously infected with TB would have an increased risk of death by other physiological causes, such as decompensation or acceleration of pre-existing diseases, or increased risk of communicable or non-communicable diseases, we defined our primary outcome as death by natural causes. This was defined as any cause of death excluding external causes (ICD-10 V01-Y98: External causes of morbidity and mortality) and causes related to TB (ICD-10 A15-A19) and HIV (ICD-10 B20-B24). Given the well-established link between HIV and TB[45], and to prevent overestimation of mortality directly due to active TB itself, excluding deaths directly attributed to TB and HIV allows for a clearer assessment of TB's broader adverse effects, potentially indirect or long-term, on health.

Secondary outcomes included all-cause mortality and cause-specific mortality defined by ICD-10 codes from causes with higher proportion of deaths in previous studies[4,8,10], specifically (i) Cardiovascular causes (Chapter IX), and blocks: Ischemic heart diseases (I20-I25), and Cerebrovascular diseases (I60-I69); (ii) Metabolic causes (Chapter IV); (iii) Respiratory System (Chapter X); (iv) Cancer (Chapter II), and blocks: Malignant neoplasms of respiratory and intrathoracic organs (C30-C39), and Malignant neoplasms of digestive organs (C15-C26); and (v) External Deaths (Chapter XX), and blocks: Accidents (V01-X59), and Assaults (X85-Y09).

### Study population and statistical analysis

Exposed individuals were exactly matched to an unexposed individual (participants not linked to a TB record) on the day of diagnosis of TB (diagnosis cohort) or date of treatment completion (treated cohort). Exact matching was performed without replacement on year of birth (in 5-year bins), sex, race or ethnicity, city of residence, household location, household water supply type, household material, year of registration in the CadÚnico (in 3-year bins) and household crowding. When multiple unexposed participants were available for a single exposed participant, one was chosen at random. Matching by these factors provided demonstrable control of bias in a previous study[46]. Unexposed participants matched on a given day who acquired TB on a subsequent date became exposed and could be matched to a new unexposed participant. In this case, they could contribute first as unexposed and later as exposed.

We excluded individuals (i) aged 100 years or older at CadÚnico registration, (ii) diagnosed with TB before the CadÚnico registration, (iii) with missing data in one of the variables used in the matching (Supplementary Table 1), (iv) data inconsistencies in date of death before diagnosis date or date of death before date of CadÚnico registration, (v) people experiencing homelessness due to the impossibility of assessing variables related to the household and (vi) participants with a TB diagnosis recorded as relapse or retreatment. For the treated cohort, we only included TB cases that were classified as "treatment success" in SINAN-TB. To remove records with inconsistent treatment length, we also excluded individuals whose treatment completion date was missing, less than 138 days after the notification date or more than 2 years after the notification date. We used this cutoff considering the treatment duration of 6-12 months, as recommended by the Ministry of Health in Brazil, depending on the clinical presentation of TB (for example, pulmonary x osseous-articular x meningoencephalic), the maximum of 2 years was chosen to allow for delays in the start of treatment after diagnosis[42].

For the diagnosed cohort, each matched pair was followed up from the matching date (that is, date of TB diagnosis for the exposed individual) until the earliest of the following events: death, or 31 December 2018 (final data collection date), diagnosis of TB in the unexposed participant (in this case, both members of the matched pair were censored). For the treated cohort, pairs' follow-up started on the day of TB treatment completion and ended on the earliest of the following events: death or 31 December 2018 (final data collection date).

We estimated the cumulative incidence function for each outcome using the Aalen-Johansen estimator, which considers the competing risk of death from other causes. For example, in the model for natural deaths, all other deaths were considered as competing causes. This estimates the total effect of TB on the cause of interest, capturing both the direct pathway by which TB affects the cause of interest and the indirect effect of TB on the competing causes[47].

We estimated marginal period-specific risks, RD, RRs and IRRs comparing the exposed group to the unexposed group for each

outcome. The period-specific intervals were demarcated on days 30, 90, 180 and 365 and yearly intervals of up to 14 years.

Subgroup analyses were conducted by sex, age at diagnosis of TB (<18, 18-59 and ≥60 years), race (White/mixed/Black), type of TB (pulmonary, extrapulmonary and extrapulmonary + pulmonary-both-), diagnosis of HIV, and diagnosis of diabetes mellitus (DM). The covariates related to the diagnosis of HIV and diabetes mellitus were extracted from the SINAN database; that is, only individuals with a diagnosis of TB have this information. In the subgroup analysis of HIV and DM, the comparison is made between individuals with a diagnosis of TB and a diagnosis of HIV or DM versus the matched unexposed participant.

We used non-parametric bootstrapping (resampling only matched pairs) with 500 iterations to calculate percentile-based 95% confidence intervals for all measures. This procedure has been proven valid when conducting matching without replacement[48]. All analyses were conducted using R, with the package survival.

### Sensitivity analysis: household contacts
To further understand how much of the excess burden could be due to shared social factors and estimate the residual confounding by unmeasured variables, we conducted an additional cohort matching TB contacts to individuals free of TB (that is, without TB diagnosis and not a TB contact). TB household contacts were eligible if they resided in the same household as a TB case and were alive at the time of the first TB case diagnosis; that is, we did not include persons born after the first TB case was diagnosed in the household. The same matching variables used in the main analysis were used. A TB contact who became a TB case afterwards was censored on the day of TB diagnosis (pair censoring). The time zero in this cohort was the date of the first TB case in the household. Each matched pair was followed up from the matching date until the earliest of the following events: death, 31 December 2018 or diagnosis of TB (in this case, both members of the matched pair were censored).

We also evaluated the comparison directly between the first TB case and the household contacts, matching only on year of birth (in 5-year bins) and sex. The time zero in this analysis was the date of the first diagnosis of a TB case in the household. Each matched pair was followed up from the matching date until the earliest of the following events: death, 31 December 2018, or diagnosis of TB in the unexposed participant (in this case, both members of the matched pair were censored)

We calculated RD, IRRs and RRs, as well as 95% CIs, similar to the main analysis.

### Sensitivity analysis: E-value
We also calculated the E-value for the RR comparing the exposed (diagnosed and treated TB groups) and the unexposed group. The E-value can be interpreted as the minimal strength of an association, on the RR scale, that an unmeasured confounder must be associated with both exposure and outcome to negate the observed exposure-outcome association.

### Protocol
The study design and statistical analysis plan were specified in advance of analyzing the data and are described in a publicly available protocol (https://github.com/csthiago/tuberculosis_death).

### Ethics
The study was approved by the ethics committees of the Instituto Gonçalo Muniz–Oswaldo Cruz Foundation (1.612.302), Salvador, Brazil.

### Reporting summary
Further information on research design is available in the Nature Portfolio Reporting Summary linked to this article.

### Data availability
The relevant data are available in the manuscript and the Supplementary Information. Raw data are available upon reasonable request to the Centro de Integração de Dados e Conhecimentos para a Saúde (CIDACS). Any person who wishes to receive authorization must: (1) be affiliated to CIDACS or be accepted as collaborators; (2) present a detailed research project together with approval by an appropriate Brazilian institutional research ethics committee; (3) provide a clear data plan restricted to the objectives of the proposed study and a summary of the analyses plan intended to guide the linkage and data extraction of the relevant set of records and variables; (4) sign terms of responsibility regarding the access and use of data; and (5) perform the analyses of datasets provided using the CIDACS data environment, a safe and secure infrastructure that provides remote access to de-identified or anonymised datasets and analysis tools. For more information: https://cidacs.bahia.fiocruz.br/.

### Code availability
The modeling in this paper used R v.4.1.1 and the tidyverse v.1.3.2, survival v.3.7-0 R packages, all of which are freely available.

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

### Acknowledgements

This study was partially supported by the National Institute of Science and Technology in Digital Health - DigiSaude-INCT (CNPq - 408775/2024-6) and Wellcome Trust (226306/Z/22/Z). V.S.B. is a Brazilian National Research Council research fellow. E.S.P. and J.M.P. acknowledge funding from the Wellcome Trust (225925/Z/22/Z to E.S.P., 305644/Z/23/Z to JMP). T.C.-S. acknowledges funding from the Royal Society (NIF\R1\231435). O.R. is funded by the Ramón y Cajal program (RYC2023-002923-C) awarded by the Spanish Ministry of Science, Innovation and Universities (MICIU/AEI/10.13039/501100011033) and by the European Social Fund Plus (ESF+). The funders had no role in study design, data collection and analysis, decision to publish or preparation of the manuscript.

### Author contributions

T.C.-S., E.S.P. and J.M.P. conceived the idea for the study. All authors contributed to the study design, with J.M.P. and T.C.-S. drafting the statistical analysis plan. T.C.-S. conducted the statistical analysis. J.M.P., C.L. and O.R. oversaw the analysis. M.L.B. and M.S. acquired the data.

T.C.-S. drafted the paper, with assistance from J.M.P. All authors critically revised the paper and approved the final version for submission.

## Competing interests

The authors declare no competing interests.

## Additional information

**Extended data** is available for this paper at https://doi.org/10.1038/s41591-026-04294-w.

**Correspondence and requests for materials** should be addressed to Thiago Cerqueira-Silva.

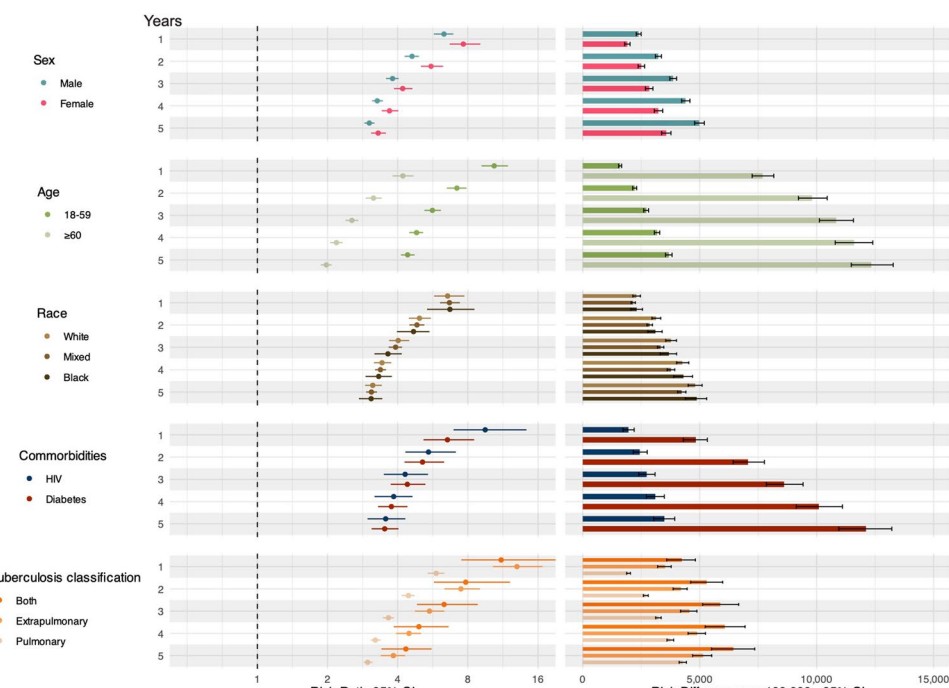

**Extended Data Fig. 1 | Stratified analysis for the diagnosed TB-exposed.**
Stratified estimated risk ratios and difference of death from natural causes (excluding external causes, HIV and TB deaths) in diagnosed TB cases compared with the unexposed group. In TB classification, "both" indicates pulmonary plus extrapulmonary. The error bars represent 95% confidence intervals. The x-axis for the risk ratio is in the log scale. The number of individuals in each group by year of follow-up is displayed in Supplementary fig. 1.

# Reporting Summary

## Statistics

For all statistical analyses, confirm that the following items are present in the figure legend, table legend, main text, or Methods section.

| n/a | Confirmed | |
|---|---|---|
| ☐ | ☒ | The exact sample size (*n*) for each experimental group/condition, given as a discrete number and unit of measurement |
| ☒ | ☐ | A statement on whether measurements were taken from distinct samples or whether the same sample was measured repeatedly |
| ☒ | ☐ | The statistical test(s) used AND whether they are one- or two-sided<br>*Only common tests should be described solely by name; describe more complex techniques in the Methods section.* |
| ☐ | ☒ | A description of all covariates tested |
| ☐ | ☒ | A description of any assumptions or corrections, such as tests of normality and adjustment for multiple comparisons |
| ☐ | ☒ | A full description of the statistical parameters including central tendency (e.g. means) or other basic estimates (e.g. regression coefficient) AND variation (e.g. standard deviation) or associated estimates of uncertainty (e.g. confidence intervals) |
| ☒ | ☐ | For null hypothesis testing, the test statistic (e.g. *F*, *t*, *r*) with confidence intervals, effect sizes, degrees of freedom and *P* value noted<br>*Give P values as exact values whenever suitable.* |
| ☒ | ☐ | For Bayesian analysis, information on the choice of priors and Markov chain Monte Carlo settings |
| ☒ | ☐ | For hierarchical and complex designs, identification of the appropriate level for tests and full reporting of outcomes |
| ☒ | ☐ | Estimates of effect sizes (e.g. Cohen's *d*, Pearson's *r*), indicating how they were calculated |

*Our web collection on statistics for biologists contains articles on many of the points above.*

## Software and code

Policy information about availability of computer code

| | |
|---|---|
| Data collection | R version 4.1.1 |
| Data analysis | R version 4.1.1; packages: survival; tidyverse; EValue |

For manuscripts utilizing custom algorithms or software that are central to the research but not yet described in published literature, software must be made available to editors and reviewers. We strongly encourage code deposition in a community repository (e.g. GitHub). See the Nature Portfolio guidelines for submitting code & software for further information.

## Data

Policy information about availability of data

All manuscripts must include a data availability statement. This statement should provide the following information, where applicable:
- Accession codes, unique identifiers, or web links for publicly available datasets
- A description of any restrictions on data availability
- For clinical datasets or third party data, please ensure that the statement adheres to our policy

The relevant data are available in the manuscript and the appendix. Raw data are available upon reasonable request to the Centro de Integração de Dados e Conhecimentos para a Saúde (CIDACS). Any person who wishes to receive authorisation must: (1) be affiliated to CIDACS or be accepted as collaborators; (2) present a detailed research project together with approval by an appropriate Brazilian institutional research ethics committee; (3) provide a clear data plan restricted to the objectives of the proposed study and a summary of the analyses plan intended to guide the linkage and data extraction of the relevant set of

records and variables; (4) sign terms of responsibility regarding the access and use of data; and (5) perform the analyses of datasets provided using the CIDACS data environment, a safe and secure infrastructure that provides remote access to de-identified or anonymised datasets and analysis tools. For more information: https://cidacs.bahia.fiocruz.br/

# Research involving human participants, their data, or biological material

Policy information about studies with human participants or human data. See also policy information about sex, gender (identity/presentation), and sexual orientation and race, ethnicity and racism.

| | |
|---|---|
| Reporting on sex and gender | We used sex (biological attribute) when describing the sex of study population. We also conducted stratified analysis by sex |
| Reporting on race, ethnicity, or other socially relevant groupings | We used race (classified in accordance to the IBGE - Instituto Brasileiro de Geografia e Estatística), white/black/mixed/asian-yellow/indigenous. This variable is self-reported. Subgroup analysis by race/ethnicity was conducted to assess structural racism. |
| Population characteristics | All the information avaliable in the database is presented in the Table 1, Supplementary Table 1,2,3 of the manuscrirpt |
| Recruitment | This is a observational study using data from national administrative databases |
| Ethics oversight | The study was approved by the ethics committees of the Instituto Gonçalo Muniz–Oswaldo Cruz Foundation (1.612.302), Salvador, Brazil. |

Note that full information on the approval of the study protocol must also be provided in the manuscript.

# Field-specific reporting

Please select the one below that is the best fit for your research. If you are not sure, read the appropriate sections before making your selection.

☒ Life sciences ☐ Behavioural & social sciences ☐ Ecological, evolutionary & environmental sciences

For a reference copy of the document with all sections, see nature.com/documents/nr-reporting-summary-flat.pdf

# Life sciences study design

All studies must disclose on these points even when the disclosure is negative.

| | |
|---|---|
| Sample size | Given that the sample sizes were based on national linked datasets; we only excluded individuals that do not fit the eligibility criteria. Resulting in 371,842 participants. |
| Data exclusions | We excluded individuals (i) aged 100 years or older at CadÚnico registration, (ii) diagnosed with tuberculosis before the CadÚnico registration, (iii) with missing data in one of the variables used in the matching (Supplementary Table 1), (iv) data inconsistencies in date of death before diagnosis date or date of death before date of CadÚnico registration, (v) people experiencing homelessness due to the impossibility of assessing variables related to the household; (vi) participants with a TB diagnosis recorded as relapse or retreatment. For the treated cohort, we only included TB cases that were classified as "cured" in SINAN-TB. To remove records with inconsistent treatment length, we also excluded individuals whose treatment completion date was missing, less than 138 days after the notification date or more than 2 years after the notification date. We used this cut-off considering the treatment duration of 6 - 12 months, as recommended by the Ministry of Health in Brazil, depending on the type of Tuberculosis, the maximum of two years was chosen to allow for delays in the start of treatment after diagnosis. |
| Replication | Not applicable |
| Randomization | Not applicable (observational study) |
| Blinding | Not applicable (observational study) |

# Reporting for specific materials, systems and methods

We require information from authors about some types of materials, experimental systems and methods used in many studies. Here, indicate whether each material, system or method listed is relevant to your study. If you are not sure if a list item applies to your research, read the appropriate section before selecting a response.

## Materials & experimental systems

| n/a | Involved in the study |
|---|---|
| ☒ | ☐ Antibodies |
| ☒ | ☐ Eukaryotic cell lines |
| ☒ | ☐ Palaeontology and archaeology |
| ☒ | ☐ Animals and other organisms |
| ☒ | ☐ Clinical data |
| ☒ | ☐ Dual use research of concern |
| ☒ | ☐ Plants |

## Methods

| n/a | Involved in the study |
|---|---|
| ☒ | ☐ ChIP-seq |
| ☒ | ☐ Flow cytometry |
| ☒ | ☐ MRI-based neuroimaging |

## Plants

| | |
|---|---|
| Seed stocks | NA |
| Novel plant genotypes | NA |
| Authentication | NA |

