## [Peer Review File · Nature Medicine]

Long-term risk of death after Tuberculosis diagnosis and treatment

Corresponding Author: Dr Thiago Cerqueira-Silva

Version 0:

Reviewer comments:

Reviewer #1

(Remarks to the Author)
NMED-A143665

A. Summary of the key results

This study used linked records from Brazil's nationwide databases of socioeconomic welfare programs, TB registries, and mortality system to assess the relationship between tuberculosis and risk of death during 2004-2018. Data assembled were used to analyze a cohort of patients diagnosed with TB and a separate cohort of patients who completed TB treatment. The risk of all-cause mortality and cause-specific mortality was compared in the two TB cohorts to a matched individual without a TB record. As expected, the rates of all-cause and cause-specific mortality were higher in the TB-diagnosed and TB-treated cohorts compared to matched controls. In the TB-diagnosed cohort, the increased risk of "natural" mortality was greatest within 30 days of TB diagnosis. In the TB-treated cohort, the relative risk of mortality remained between 2-3 times that of controls throughout the follow up. Broadly, the cause-specific and stratified analyses of mortality were not different than the primary analyses.

B. Originality and significance: if not novel, please include reference

The objectives of this study, and its key result, do not differ substantially from the current systematic review paradigm that is used to estimate the relationship between TB and all-cause mortality (Romanowski, et al, Lancet Infect Diseases, 2019 PMID 31324519). Nonetheless, this study by Cerqueira-Silva et. al. does have substantial methodologic advantages and rigor compared to previous studies with the same objective (to estimate the relationship between TB and long-term mortality risk). This study's greatest strength over previous work is that it analyzes a very large population of patients with history of TB who are precisely matched to controls without TB and followed for an extensive period of time. Additionally, the ability to compare relative and absolute differences in mortality risk during specific periods of time after TB (30 days, 1-year, 5-years, 10-years, 14-years) is an important strength and improvement over previous work. The cause-specific analyses of mortality would be another substantial improvement, however, misclassification of cause of death due to reliance on ICD codes in this study is likely an important source of bias.

C. Data & methodology: validity of approach, quality of data, quality of presentation

The primary outcome for this study is deaths by natural causes, defined by any cause of death excluding external causes. This definition of natural vs. external is unclear and inhibits the interpretation of results. Although the authors provide a reference to the 10th edition of ICD, the lack of details in the Methods does not provide the reader with sufficient information to clearly understand the rationale for this outcome. The justification for not using all-cause mortality as the primary outcome is not explained.

The TB diagnosed cohort includes participants with an indication of TB disease in the Brazil national notifiable disease system. This likely includes clinical cases or cases without culture confirmation. Whether the TB diagnosed cohort includes non-culture confirmed cases needs to be clarified and the proportion that do not have culture confirmation reported. A subgroup analysis of those with culture confirmed TB would be informative as many of the clinical cases are likely misclassified as TB disease.

The TB treated cohort includes participants with two negative smear results or do not have recorded evidence of treatment failure. Similar to information about culture confirmation for the TB diagnosed cohort, this TB treated cohort analysis would

be more robust if limited to those with confirmed smear conversion AND treatment success. No evidence of treatment failure is subject to misclassification and likely includes cases that were not successfully treated.

Additional information is needed to describe how TB relapse cases were treated. Those with TB relapse appear to be excluded based on Figure 1, but this is not explained in the Methods. Exclusion of TB relapse cases will substantially underestimate the overall mortality burden attributable to TB.

Figure 1 indicates there are 209,598 in the tuberculosis diagnosed group, 17,370 with relapse, and 83,609 with missing TB treatment completion information. This suggests an approximate TB successful treatment rate of ~52%. This proportion is significantly lower than Brazilian national estimates for treatment success rates, even among relapse cases. Whether these data are generalizable to the national population is uncertain.

D. Appropriate use of statistics and treatment of uncertainties

The authors' use of statistics is overall acceptable. However, some details and the rationale for mixed use of measures of associations is unclear. The analysis reports both absolute and relative measures of association which is a strength. It is unclear why the authors have not used survival analyses and proportional hazards models, the standard approach for the competing risk and censored data included in this analysis.

For example, additional information is needed to explain how participants who fell out of individual risk sets (due to events other than the primary outcomes) are handled in future risk intervals. There is no definition or explanation of participants who would be considered censored for loss to follow up. The authors do not explain how participants who fall out of the CadUnico database are treated—presumably people move out of the country or cannot be followed continuously for various reasons. It is unclear whether the incidence intervals were only calculated at the time points of 30 days, 1-year, 5-years, 10-years, 14-years and why this would be preferred over an instantaneous rate ratio (hazard ratio). The authors do not adequately justify the use of different measures of association, they report RD, IRR, and RR.

Competing risk cumulative incidence functions for cause-specific mortality would be useful to present.

E. Conclusions: robustness, validity, reliability

Overall there are no major concerns regarding this manuscript's validity or reliability. The principal concerns related to bias due to misclassification, the analytic approach, and bias due to confounding are described above and below.

F. Suggested improvements: experiments, data for possible revision

A major limitation of this manuscript is that the analysis did not take into consideration prevalence of key morbidities prior to date of TB diagnosis. Presumably the authors should be able to determine if participants with a TB diagnosis had ICD indications of major diseases (diabetes, cardiovascular diseases, respiratory disease, cancers) before the date of TB diagnosis. This information should also be available for controls prior to an index date based on matched age. Accounting for prior diagnoses of major diseases associated with TB morbidity and mortality would improve the robustness and validity of this analysis.

While there is no way to account for all unmeasured confounders, a bias analysis for unmeasured confounding would be useful. For example, one key confounder not accounted for in this analysis is TB infection (in contrast to TB disease). TB infection is associated with increased risk of multiple non-communicable diseases and mortality risk. The authors could perform a bias analysis due to unmeasured confounding from TB infection or more broadly calculate an e-value to estimate the potential confounding relationships necessary to explain away the main study results.

G. References: appropriate credit to previous work?

References are appropriate.

H. Clarity and context: lucidity of abstract/summary, appropriateness of abstract, introduction and conclusions

Abstract is appropriate.

The Figures and Tables are difficult to read.

Reviewer #2

(Remarks to the Author)

Thank you for the possibility to review this manuscript. The authors have described an increased risk of death after diagnosis and treatment completion of tuberculosis. While this is not a new finding, the authors have assessed several new and interesting aspects. Firstly, they have been able to match controls much closer to cases when it comes to socio-economic aspects and also assessed the effect of socio-economic factors itself on the risk of death; thereby showing that the increased risk of death after tuberculosis is not just based on poorer socio-economic circumstances that often underlie tuberculosis.

Secondly, they have distinguished between patients diagnosed with TB and patients completing TB treatment, showing that the increased risk of death is not only in those who have not been successfully treated for TB.

The methods have mostly been clearly described and the manuscript is easy to read. The findings are clearly presented and seem straightforward. It is almost a pity that the authors try to explain their findings by adding in some immunological details (line 410), which seem to be out of place in the context of the manuscript.

I have a few minor comments / questions:

Line 53-55: WHO end TB strategy: these sentences read a bit weird as we are already halfway through 2025; are there any updated numbers available?

Line 105: I would like to see some more details about how the databases were linked. Also, I do not understand what the sensitivity and specificity refer to.

Figure 2: is the figure legend complete?

Line 357: why is this cohort so much younger?

There are some spelling / text errors (eg. line 408, line 443).

Reviewer #3

(Remarks to the Author)

Manuscript Title: Long-term risk of death after Tuberculosis diagnosis and treatment in Brazil: a nationwide longitudinal study using linked routine data between 2004 and 2018

Journal: Nature Medicine

Dear authors,

I have carefully reviewed the manuscript. The study addresses an important question regarding long-term mortality risk associated with tuberculosis, and the authors leverage a large administrative dataset with detailed socioeconomic information. The overall contribution is potentially valuable for the field. I would like to acknowledge the considerable effort the authors have invested in assembling and analysing this dataset; the exhaustive analyses presented reflect both the importance of the topic and the authors' strong commitment to producing meaningful results.

General Assessment of the Study

Data Quality: The study is based on secondary data sources. While this inevitably implies the possibility of some missing information or unmeasured variables, the available data appear to be of high quality, as reported by the authors. The dataset and the methods used for extraction and analysis seem reliable and transparent, and a brief clarification of data completeness could further help readers appreciate the strengths of the information.

Level of Support for the Conclusions: The findings are compelling, but certain clarifications and additional analyses (or discussions of limitations) could make the conclusions more robust. This includes a clearer specification of the study design, a more detailed handling of comorbidities and sociodemographic variables, and an explicit mention of unmeasured factors such as latent TB infection.

Significance of the Results: The results are relevant, interesting, and necessary to advance understanding of the long-term impact of tuberculosis. By refining some analytic choices and clarifying design considerations, the study has strong potential to move the field forward.

Elements of a Reviewer Report

Key Results: The results are highly interesting and contribute valuable information to the field. However, despite the detailed analysis, some findings do not appear to be fully supported by the presented evidence, and a stronger alignment between results and conclusions would improve the manuscript.

Validity: The data and analyses are generally reliable, though some methodological clarifications (study design, handling of comorbidities, and possible misclassification) would enhance robustness. None of these issues preclude publication but should be openly addressed.

Significance: The conclusions have the potential to influence understanding of long-term TB outcomes. Greater clarity regarding confounding factors and the role of latent infection would strengthen the study's impact.

Data and Methodology: The dataset is rich and well-documented, though further clarification on completeness, coding of comorbidities, and supplementary analyses would help readers assess reproducibility.

Analytical Approach: The use of stratified analyses provides granularity, but it introduces potential challenges related to multiplicity. Considering alternative multivariable approaches could complement the findings.

Suggested Improvements: Additional clarifications, some targeted sensitivity analyses, and a more explicit discussion of limitations would be valuable to reinforce the conclusions within the current scope of the manuscript.

Clarity and Context: The text is generally clear, but the discussion would benefit from additional framing in the context of existing literature on long-term TB outcomes.

References: The cited literature is relevant, but expanding on prior studies of mortality in household contacts and latent TB would better situate the work within the broader field.

Major Comments

1. Study Design and Terminology

Comment: The manuscript refers to the study as a cohort design. However, the matching procedures on multiple baseline covariates suggest features of a matched cohort or even a nested case–control/case–cohort framework. This distinction influences how results are interpreted.

Suggestion: It may help readers if the authors clarify explicitly how they conceptualize the design (matched cohort vs. nested case–control) and briefly note its implications for analysis. Alternative descriptions may also be valid, but additional clarity would enhance transparency.

2. Identification of Prior TB Episodes

Comment: It is unclear whether the dataset allows identification of previous TB episodes before study entry. If not, there may be a risk of misclassifying recurrent TB as incident, which could influence mortality estimates.

Suggestion: If feasible, specify how prior TB was assessed. If this information is not available, a short discussion of the potential impact and its direction would help readers interpret the findings.

3. Sociodemographic and Household Variables

Comment: The introduction highlights sociodemographic factors, yet the analyses focus mainly on sex, age, and ethnicity. Other socioeconomic and household variables (e.g., housing conditions, household size, registration year) may also provide additional insights.

Suggestion: The authors could consider incorporating these variables more explicitly, perhaps in multivariable models or interaction terms. If this is not possible, a brief acknowledgement of the limitation would guide readers in framing the results.

4. Comorbidities and Temporal Dynamics

Comment: It is not clear whether comorbidities (such as diabetes or HIV) were measured dynamically during follow-up or only at baseline. This matters because comorbidities may arise after TB diagnosis and affect mortality risk differently from pre-existing conditions.

Suggestion: If longitudinal information is available, distinguishing pre-existing from incident comorbidities could be informative. If not, a short clarification in the methods and a note in the discussion would be helpful.

5. Multiplicity and Stratification

Comment: The number of stratified analyses (by sex, age, ICD chapters, ICD blocks) provides detail but increases the risk of multiplicity and small sample sizes in some strata.

Suggestion: An alternative could be to employ multivariable competing-risk models (e.g., Fine–Gray regression) that handle several covariates simultaneously. This might reduce the multiplicity burden while maintaining interpretability. If the authors prefer the current approach, a brief discussion of its trade-offs would still be valuable.

6. Contacts as a Separate Group

Comment: Household contacts are analysed separately, but direct comparisons between TB cases and contacts are not shown. This may limit the ability to distinguish effects of TB disease from those of the shared environment.

Suggestion: If possible, including direct comparisons of cases and contacts (using matching or propensity score methods) would provide further insights. Alternatively, a short explanation of why this was not pursued would guide the reader.

7. Latent TB Infection, Reactivation, and Reinfection

Comment: The study does not directly measure LTBI or distinguish reactivation from reinfection, yet these factors are central for interpreting mortality in contacts.

Suggestion: It could strengthen the discussion to acknowledge this explicitly and, if feasible, include a sensitivity or bias analysis using plausible LTBI prevalence estimates. If this is not possible, even a qualitative discussion would help contextualize the findings.

Minor Comments

The supplementary tables are extensive. The authors may consider consolidating some material or highlighting the most essential comparisons in the main text to improve readability. Consistent terminology for “diagnosed cohort,” “treated cohort,” and “household contacts” would aid clarity. Including E-values or another quantitative measure of robustness could further support interpretation. Finally, a note on whether household clustering was addressed (e.g., through multilevel modelling) would reassure readers about potential shared vulnerability.

Reviewer #4

(Remarks to the Author)

This manuscript presents significant findings; although retrospective in nature, the information provided is valuable in understanding mortality related to tuberculosis diagnosis, as well as outcomes following completion of treatment and bacteriological cure.

The introduction is clear and concise, and the goal of the study is established.

Methods: Data from three separate registries in Brazil were cross-referenced, and two cohorts were extracted from these combined databases: Diagnosed Cohort and treated cohort

CadUnico: registry for low income families, how this is defined?

SINAN: TB registry

It is important to include how TB diagnosis is made.

All included cases were microbiologically confirmed or suspected cases were also included?

For the treated cohort only patients with positive outcome were included?

For the treated cohort, there is no minimum number of doses??

Patients with only one TB event were included? Relapse cases were not included? This is important information as relapse has an impact on mortality.

All treated cases received the same regimen?

SIM: mortality information

The inclusion criteria and matched controls are explicitly described.

Subgroup analysis was conducted by sex, age at diagnosis, race, type of TB Pulmonary, extrapulmonary and pulmonary + extrapulmonary. It is essential to specify the number of CNS-TB patients included, as this is a severe form of tuberculosis and associated with increased mortality and morbidity.

Were all included cases drug sensitive? No drug-resistant cases were included?. This distinction is relevant because drug-resistant tuberculosis typically presents more treatment challenges and is associated with higher mortality rates. Brazil also reports a significant number of drug-resistant TB cases.

Household contacts cohort: How many individuals were screened for latent tuberculosis, and how many received preventive treatment? Also important for mortality and morbidity

The diagnosed cohort was followed from diagnosis until death or December 31, 2018; the treated cohort was followed from treatment completion until death or December 31, 2018. Was there no minimum follow-up period?

Does the study have ethical approval?

Results:

The median follow-up duration differs between the diagnosed and treated cohorts and the control groups. Specifically, the controls have a shorter follow-up period compared to the diagnosed and/or treated cohorts. This discrepancy may be attributable to higher mortality rates among the control group?

Regarding cause-specific mortality, it is important to highlight which causes are more prevalent within each cohort.

“Participants with diabetes mellitus had a similar RR to those with HIV in the diagnosed cohort and a higher RR in the treated cohort. Notably, the absolute excess mortality (risk difference) in TB patients with DM was substantially larger, exceeding double the RD observed in TB patients with HIV”.

This paragraph refers to comparisons made with patients diagnosed and treated for TB, rather than with control groups, as the control groups do not specify the presence of DM or HIV.

The results indicate an association between TB diagnosis and TB-treated cases with cancer, especially gastrointestinal cancer.

Patients are matched with controls based on age, sex, and socioeconomic status, rather than comorbidities. It is important to include (if this information is available) that no patient had a prior diagnosis of cancer.

DISCUSSION:

Line 389: “Relying solely on time after diagnosis can group individuals who abandoned treatment, experienced treatment failure, who are classified as drug-resistant TB, or were lost to follow-up with those who completed treatment, likely biasing the results”. This sentence seems incomplete

This section details the number of drug-resistant cases included. It is necessary to incorporate this information in the results section and clarify the type of resistance, as outcomes may vary accordingly.

Limitations of the study are clearly stated

Another limitation of the study is the absence of control group for DM and HIV cases

Conclusions not so clear, should be rephrased

Figures and tables:

Figure 2: “Yearly incidence rate ratios from natural causes “ it is missing one word maybe death

Version 1:

Reviewer comments:

Reviewer #1

(Remarks to the Author)

The authors have adequately responded to my concerns.

Reviewer #3

(Remarks to the Author)

Dear Authors,

I have carefully reviewed the revised version of the manuscript. Taking into account both my previous comments and those raised by the other reviewers, I believe that this new version is substantially strengthened. The study is now more clearly framed, its limitations are more explicitly acknowledged, and the conclusions are appropriately supported by the presented results.

I appreciate the authors' thorough and thoughtful responses to my questions and observations. In my view, the manuscript in its current form meets the methodological standards and scientific quality required for publication.

That said, I would like to offer two final, minor suggestions that could further strengthen the interpretation of the results. I consider these optional and leave their inclusion or modification entirely at the authors' discretion.

First, regarding latent tuberculosis infection (LTBI) and residual confounding, I suggest considering the inclusion of the following statement in the Limitations section, to clarify the interpretation of the estimated effects:

"Thus, our results reflect excess mortality following a TB disease episode, not the isolated biological effect of TB, as LTBI and related pre-existing social and health vulnerabilities are unmeasured."

Second, to further guide the interpretation of the subgroup and cause-specific analyses and to address concerns related to multiplicity, I suggest adding a brief clarification in the Limitations section, such as:

"Accordingly, subgroup and cause-specific estimates should be interpreted as descriptive assessments of heterogeneity and patterning rather than as confirmatory evidence for differential effects."

Reviewer #4

(Remarks to the Author)

All queries have been addressed appropriately.

While the authors recognize several limitations within the study, the insights and findings presented still hold significant relevance and value.

Responses to Reviewers' Comments:

Reviewer #1 (Remarks to the Author):

NMED-A143665

R1-P1) A. Summary of the key results

This study used linked records from Brazil's nationwide databases of socioeconomic welfare programs, TB registries, and mortality system to assess the relationship between tuberculosis and risk of death during 2004-2018. Data assembled were used to analyze a cohort of patients diagnosed with TB and a separate cohort of patients who completed TB treatment. The risk of all-cause mortality and cause-specific mortality was compared in the two TB cohorts to a matched individual without a TB record. As expected, the rates of all-cause and cause-specific mortality were higher in the TB-diagnosed and TB-treated cohorts compared to matched controls. In the TB-diagnosed cohort, the increased risk of "natural" mortality was greatest within 30 days of TB diagnosis. In the TB-treated cohort, the relative risk of mortality remained between 2-3 times that of controls throughout the follow up. Broadly, the cause-specific and stratified analyses of mortality were not different than the primary analyses.

R: Thank you for your time to review our work.

R1-P2) B. Originality and significance: if not novel, please include reference

The objectives of this study, and its key result, do not differ substantially from the current systematic review paradigm that is used to estimate the relationship between TB and all-cause mortality (Romanowski, et al, Lancet Infect Diseases, 2019 PMID 31324519). Nonetheless, this study by Cerqueira-Silva et. al. does have substantial methodologic advantages and rigor compared to previous studies with the same objective (to estimate the relationship between TB and long-term mortality risk). This study's greatest strength over previous work is that it analyzes a very large population of patients with history of TB who are precisely matched to controls without TB and followed for an extensive period of time. Additionally, the ability to compare relative and absolute differences in mortality risk during specific periods of time after TB (30 days, 1-year, 5-years, 10-years, 14-years) is an important strength and improvement over previous work. The cause-specific analyses of mortality would be another substantial improvement, however, misclassification of cause of death due to reliance on ICD codes in this study is likely an important source of bias.

R: We thank the reviewer for the comments. Regarding the use of ICD-10 codes, in Brazil, death certificate information is collected by trained mortality coders (after appropriate training according to the World Health Organization standards). They work in statistical regional offices of the Ministry of Health and use mortality-coding software specifically designed for this purpose. This practice has remained constant since 1996 (when ICD-10 was introduced in Brazil), with studies evaluating specific causes of death finding similar results to countries using other methods of classification of death [1].

Also most deaths classified with a garbage code occur within the same ICD groups (i.e., in 2015, 27.5% of garbage codes occurred within the same chapter) [2], as our analysis uses ICD-10 chapters or blocks, it is unlikely to bias the results. Although we acknowledge the presence of misclassification, we believe it is unlikely to have strongly affected our estimates as we looked both at mortality after TB by chapters and specific causes within those.

We have included as limitation the reliance on ICD codes: *“Eighth, we relied on ICD-10th codes from the mortality information system to classify the underlying cause of death, while there is no comprehensive study evaluating the quality of this codification in Brazil. The quality of the death certificate information assessed by the percentage of garbage codes has improved nationwide, with a reduction of up to 90% of garbage codes between 2000 and 2015.”*⁴⁶

[1] Taniguchi, Leandro U., et al. "Sepsis-related deaths in Brazil: an analysis of the national mortality registry from 2002 to 2010." *Critical Care* 18.6 (2014): 608.

[2] Teixeira, Renato Azeredo, et al. "Quality of cause-of-death data in Brazil: Garbage codes among registered deaths in 2000 and 2015." *Revista Brasileira de Epidemiologia* 22 (2019): e19002-supl.

R1-P3) C. Data & methodology: validity of approach, quality of data, quality of presentation

The primary outcome for this study is deaths by natural causes, defined by any cause of death excluding external causes. This definition of natural vs. external is unclear and inhibits the interpretation of results. Although the authors provide a reference to the 10th edition of ICD, the lack of details in the Methods does not provide the reader with sufficient information to clearly understand the rationale

for this outcome. The justification for not using all-cause mortality as the primary outcome is not explained.

R: Thank you for raising this important point. All of our outcomes were defined using ICD-10th codes, as every death in Brazil is assigned an underlying cause based on the ICD-10th codes. The natural causes are all causes excluding the Chapter XX of ICD-10th (“External causes of morbidity and mortality”). We have included the rationale for the use of this outcome in the methods section: *“To test our hypothesis that people previously infected with TB would have an increased risk of death by other physiological causes, such as decompensation or acceleration of pre-existing diseases, or increased risk of communicable or non-communicable diseases, we defined our primary outcome as death by natural causes. This was defined as any cause of death excluding external causes (International Classification of Diseases, 10th edition (ICD-10 V01-Y98: External causes of morbidity and mortality) and causes related to Tuberculosis (ICD-10 A15-A19) and HIV (ICD-10 B20-B24). Given the well-established link between HIV and TB,14 and to prevent overestimation of mortality directly due to active TB itself, excluding deaths directly attributed to TB and HIV allows for a clearer assessment of TB’s broader adverse effects, potentially indirect or long-term, on health.”*

We have taken this decision because the use of all-cause mortality would artificially increase the risk of death in the tuberculosis group, as it would include all deaths directly attributable to TB, while the control group would not have deaths in this category, as by definition they are TB-free. We also excluded deaths attributable to HIV, due to the strong association between TB and HIV, in this case the TB exposed group would also have much larger deaths from HIV than the non-exposed. This can be seen in cumulative incidence function curves (sFigure 2). We have now included in the Supplementary material, the cumulative incidence functions for cause-specific, including HIV and TB (see Supplementary Figure 2 and 4). In those plots it is possible to see that the group diagnosed with TB has a disproportionate number of deaths attributed to TB or HIV. The rationale for external causes of death is that those deaths are strongly associated with socioeconomic factors/vulnerability and not directly related to Tuberculosis.

This was the main reason to not treat all-cause mortality as the main outcome, however all measures (Risk, risk ratio, risk difference and IRR) are also provided for all-cause mortality allowing for direct comparison of previous studies using all-cause mortality.

Cumulative incidence for the diagnosed cohort:

Supplementary Figure 2. Cumulative incidence curves of cause-specific mortality in the diagnosed tuberculosis cases and the unexposed control group. a) HIV and TB deaths,

b) Cancer, endocrine, respiratory and cardiovascular deaths, c) Natural deaths (i.e., deaths excluding HIV/TB and external causes)

R1-P4) The TB diagnosed cohort includes participants with an indication of TB disease in the Brazil national notifiable disease system. This likely includes clinical cases or cases without culture confirmation. Whether the TB diagnosed cohort includes non-culture confirmed cases needs to be clarified and the proportion that do not have culture confirmation reported. A subgroup analysis of those with culture confirmed TB would be informative as many of the clinical cases are likely misclassified as TB disease.

The TB treated cohort includes participants with two negative smear results or do not have recorded evidence of treatment failure. Similar to information about culture confirmation for the TB diagnosed cohort, this TB treated cohort analysis would be more robust if limited to those with confirmed smear conversion AND treatment success. No evidence of treatment failure is subject to misclassification and likely includes cases that were not successfully treated.

R: Thank you for these questions. The national TB system follows the Brazilian Ministry of Health definition of a TB case, which is based on the WHO definition [ref]. In LMIC, the TB diagnosis restricted only to TB with microbiological confirmation (smear test/Xpert MTB, culture, histopathological evidence) could bias the results to those with better access to care, those with more severe clinical presentation (because of better access and comprehensive evaluation), as well as to those more commonly with microbiological confirmation, such as Pulmonary TB. We kindly disagree that the clinical “only” TB is likely other diseases. The Brazilian notification system actively looks for differential diagnosis, and even if a notification is done, but afterwards the diagnosis of TB is discarded, this notified case is not considered TB anymore.

Regarding culture per se, due to the high cost, length to process the results (more than one month) and required infrastructure, TB culture is prioritised for use in retreatment, high-risk TB groups (e.g., HIV positive, people experiencing homelessness, drug users, among others), and contacts of drug-resistant TB. [1]

Xpert MTB/RIF has been increasingly used in high-burden cities but has not been introduced in the entire country yet. Limiting the analysis to cases with culture would induce substantial selection bias to individuals, not only to high-risk groups, but to individuals living in large cities that have better healthcare structures. As the guidelines of the Brazilian Thoracic Association in 2009, recommended culture only in specific cases, such as negative bacilloscopic, co-infection with HIV, suspicion of resistance, etc.[2] Last, it is needed to consider the Brazilian scenario (high burden TB, with limited infrastructure for laboratory diagnosis-culture)).

We have included it the limitation section:

“Fourth, only 15% of the TB cases had sputum culture positive, which is considered the gold standard for the diagnosis of TB. This proportion reflects the Brazilian guidelines, which prioritises sputum culture for suspected TB cases that present a negative bacilloscopy, and specific cases such as suspected cases of resistance, retreatment cases, in prison populations.⁴² In addition, due to the required infrastructure and time for diagnosis, the use of sputum culture as primary diagnosis for TB remains limited in LMICs.^{43,44} The inclusion of non confirmed TB patients in the TB group likely underestimate the risk in this group.”

However, it is worth noticing that 69% of our diagnosis cohort was positive at least on laboratory diagnosis test, either by culture (16%) or histopathology (10%) or bacilloscopic (56%) or molecular test (10%); and over 93% confirmed by radiographic or

any laboratory diagnosis (See Table 1 and supplementary table 2). We have included this as limitation:

Finally, we did not include treatment failure in the treated cohort, and only those who completed the treatment and were TB-free. We have now clarified that in the methods section:

“ For the treated cohort, we only included TB cases that were classified as “cured” in SINAN-TB. To remove records with inconsistent treatment length, we also excluded individuals whose treatment completion date was missing, less than 138 days after the notification date or more than 2 years after the notification date. We used this cut-off considering the treatment duration of 6 - 12 months, as recommended by the Ministry of Health in Brazil, depending on the clinical presentation of TB (e.g., pulmonary x osseous-articular x meningoencephalic), the maximum of two years was chosen to allow for delays in the start of treatment after diagnosis.¹⁰”

[1] Jabbour, Elias, et al. "Estimated costs of tuberculosis services in Brazil, 2023." *BMJ Open Respiratory Research* 12.1 (2025).

[2] Conde, Marcus Barreto, et al. "III Brazilian thoracic association guidelines on tuberculosis." *Jornal Brasileiro de Pneumologia* 35 (2009): 1018-1048.

R1-P5) Additional information is needed to describe how TB relapse cases were treated. Those with TB relapse appear to be excluded based on Figure 1, but this is not explained in the Methods. Exclusion of TB relapse cases will substantially underestimate the overall mortality burden attributable to TB.

Figure 1 indicates there are 209,598 in the tuberculosis diagnosed group, 17,370 with relapse, and 83,609 with missing TB treatment completion information. This suggests an approximate TB successful treatment rate of ~52%. This proportion is significantly lower than Brazilian national estimates for treatment success rates, even among relapse cases. Whether these data are generalizable to the national population is uncertain.

R: We have now fixed the wording in the flowchart, as missing TB treatment completion should be missing TB treatment completion or date inconsistencies.

The number used for our analysis can't be directly compared to TB treatment success rates, as not all individuals diagnosed will have sufficient follow-up time to assess treatment success (e.g. cases diagnosed in 2018).

In addition, we also excluded individuals with inconsistencies in the date of treatment completion (“*To remove records with inconsistent treatment length, we also excluded individuals whose treatment completion date was missing, less than 138 days after the notification date or more than 2 years after the notification date. We used this cut-off considering the treatment duration of 6 - 12 months, as recommended by the Ministry of Health in Brazil, depending on the type of Tuberculosis, the maximum of two years was chosen to allow for delays in the start of treatment after diagnosis*”).

Among the 209,598, a total of 145,379 is recorded as treatment completion (~69%) regardless of date of treatment completion, among the 209,598, a total of 27,262 TB cases were recorded in 2018. Using only the cases recorded until the end of 2017, we have 137,514/182,335 (~75%), which agrees with national estimates of treatment success rates [1]

[1]Rabahi, Marcelo Fouad, and Marcus Barreto Conte. "Decreasing trends in tuberculosis cure indicators in Brazil." *Jornal Brasileiro de Pneumologia* 50.2 (2024): e20240121.

R1-P6) D. Appropriate use of statistics and treatment of uncertainties

The authors use of statistics is overall acceptable. However, some details and the rationale for mixed use of measures of associations is unclear. The analysis reports both absolute and relative measures of association which is a strength. It is unclear why the authors have not used survival analyses and proportional hazards models, the standard approach for the competing risk and censored data included in this analysis.

For example, additional information is needed to explain how participants who fell out of individual risk sets (due to events other than the primary outcomes) are handled in future risk intervals. There is no definition or explanation of participants who would be considered censored for loss to follow up. The authors do not explain how participants who fall out of the CadUnico database are treated—presumably people move out of the country or cannot be followed continuously for various reasons.

R: We have used the Aalen-Johansen estimator (cumulative incidence function) to estimate the risks under each group.[1] This estimator handles competing risk (i.e. deaths from other outcomes). For example in the analysis of natural deaths, individuals

who died from external causes (ICD-10th chapter XX) are not in the risk set for natural deaths anymore. The standard non-parametric estimator of the cause-specific incidence function (competing risks) is the Aalen-Johansen estimator, also described as the 'multi-state version' of the Kaplan-Meier estimator.[1] The hazard ratio is a non collapse effect measure (the HR will change based on the covariates in the model), in addition it relies on the proportional hazard assumption, which has been often criticised as unrealistic, in addition to in-built selection bias (effect on time X is conditional on not dying in X-1).[2] Survival curves, risk differences and risk ratios have been proposed as alternatives measures not suffering from the same limitations [3], and are also more directly and intuitive interpretation, allowing answer questions such as "what is the risk of death at 5 years" [3] To have now added additional information in the methods section: *"We estimated the cumulative incidence function for each outcome using the Aalen-Johansen estimator, which considers the competing risk of death from other causes. For example, in the model for natural deaths, all other deaths were considered as competing causes. This estimates the total effect of TB on the cause of interest, capturing both the direct pathway by which TB affects the cause of interest and the indirect effect of TB on the competing causes.17"*

We have also included additional information in the methods section that could clarify the cohort definition *"Registration in CadÚnico was used to define our baseline study population independent of individuals updating or not the registry after, whilst exposure definition was based on the linkage with TB administrative records and follow-up based on linkage with mortality records. Therefore, from the linked dataset, we built two cohorts: one matching persons with a TB diagnosis to TB-free unexposed participants, and another matching participants who had completed TB treatment to TB-free unexposed participants"*

Regarding the outcomes, all outcomes were assessed through the national mortality system through probabilistic linkage. In this case, it is not necessary for the individual to be still registered in the CadÚnico database to have his outcome assessed, if the person is still in the country there is no loss of follow-up. The estimated completeness of the SIM (mortality system) was 98% in 2016 [4]. However, as pointed by the reviewer, it is not possible to follow-up individuals who move out of the country, we have now acknowledged this limitation in the discussion section

"Lastly, we were unable to censor individuals when they emigrated from Brazil. However, emigration rates for Brazil are low, estimated at 0.8% in 2019,⁴⁷ making it unlikely to substantially bias our estimates."

[1] Geskus, Ronald B. "Competing risks: concepts, methods, and software." *Annual Review of Statistics and Its Application* 11 (2015).

[2] Stensrud, Mats J., and Miguel A. Hernán. "Why use methods that require proportional hazards?." *American Journal of Epidemiology* 194.6 (2025): 1504-1506.

[3] Dumas, Elise, and Mats J. Stensrud. "How hazard ratios can mislead and why it matters in practice." *European Journal of Epidemiology* (2025): 1-7.

[4] Costa, Luiz Fernando Lima, et al. "Estimating completeness of national and subnational death reporting in Brazil: application of record linkage methods." *Population Health Metrics* 18.1 (2020): 22.

R1-P7) It is unclear whether the incidence intervals were only calculated at the time points of 30 days, 1-year, 5-years, 10-years, 14-years and why this would be preferred over an instantaneous rate ratio (hazard ratio). The authors do not adequately justify the use of different measures of association, they report RD, IRR, and RR.

Competing risk cumulative incidence functions for cause-specific mortality would be useful to present.

R: As described in the R1-P6, we decided to use (cumulative) risk ratios as the main effect measure, as it doesn't suffer from the multiple disadvantages of hazard ratios. First, the hazard ratio requires that the proportional hazard assumption be interpreted as a single measure over the study period. Second, the hazard ratio represents the instantaneous rate conditional on not having experienced the event, which is a selection bias mechanism. We have used RR and RD as the main effect measures recommended in the recent literature, due to the more interpretable nature of RR and RD and not suffering from the limitations from hazard ratio.[1] The RR and RD also provide absolute and relative measures, providing comprehensive evidence. This has been recommended for observational and RCT studies. [2, 3] The estimation of risks (and RD/RR) requires specific time points as no assumption of constant risk is made.[4,5] The use of specific time points (30, 90, 180, 365 and yearly intervals up to 14 years) were chosen to provide comprehensive information about the effects of TB. In the main manuscript, to reduce clutter, we restricted it to 30 days, 1-year, 5-years, 10-years, 14-years, as representative measures of short-medium-long term risks.

We have also included the yearly IRR, which provides the same information of yearly time varying hazard ratio,[6,7] to complement the information from RD/RR.

Lastly, we have now included cumulative incidence functions for the cause-specific (HIV/TB, and by ICD-10 groups) (Supplementary Figure 2 and 4)

[1] Dumas, Elise, and Mats J. Stensrud. "How hazard ratios can mislead and why it matters in practice." *European Journal of Epidemiology* (2025): 1-7.

[2] Noordzij, Marlies, et al. "Relative risk versus absolute risk: one cannot be interpreted without the other." *Nephrology Dialysis Transplantation* 32.suppl_2 (2017): ii13-ii18.

[3] <https://www.consort-spirit.org/>

[4] Stensrud, Mats J., and Miguel A. Hernán. "Why use methods that require proportional hazards?." *American Journal of Epidemiology* 194.6 (2025): 1504-1506.

[5] Dumas, Elise, and Mats J. Stensrud. "How hazard ratios can mislead and why it matters in practice." *European Journal of Epidemiology* (2025): 1-7.

[6] Cole SR, Hudgens MG, Brookhart MA, Westreich D. Risk. *Am J Epidemiol*. 2015 Feb 15;181(4):246-50. doi: 10.1093/aje/kwv001. Epub 2015 Feb 5. PMID: 25660080; PMCID: PMC4325680.

[7] Hernán, Miguel A. "The hazards of hazard ratios." *Epidemiology* 21.1 (2010): 13-15.

R1-P8) E. Conclusions: robustness, validity, reliability

Overall there are no major concerns regarding this manuscript's validity or reliability. The principal concerns related to bias due to misclassification, the analytic approach, and bias due to confounding are described above and below.

F. Suggested improvements: experiments, data for possible revision

A major limitation of this manuscript is that the analysis did not take into consideration prevalence of key morbidities prior to date of TB diagnosis. Presumably the authors should be able to determine if participants with a TB diagnosis had ICD indications of major diseases (diabetes, cardiovascular diseases, respiratory disease, cancers) before the date of TB diagnosis. This information should also be available for controls prior to an index date based on matched age. Accounting for prior diagnoses of major diseases associated with TB morbidity and mortality would improve the robustness and validity of this analysis.

R: We thank the reviewer for raising this point. As our baseline data is from an administrative database related to social programs, there is no information regarding comorbidities. We can only assess the comorbidities in the individuals notified with Tuberculosis through the SINAN notification system. We have rephrased the text in the manuscript to make it clear

“Subgroup analyses were conducted by sex, age at diagnosis of TB (<18, 18-59 and ≥60 years), race (white/mixed/black), type of TB (pulmonary, extrapulmonary and extrapulmonary + pulmonary), diagnosis of HIV, and diagnosis of diabetes mellitus (DM). The covariates related to the diagnosis of HIV and diabetes mellitus were extracted from the SINAN database; i.e., only individuals with a diagnosis of TB have this information. In the subgroup analysis of HIV and DM, the comparison is made between individuals with a diagnosis of or treated TB and a diagnosis of HIV or DM versus the matched unexposed participant.”

R1-P9) While there is no way to account for all unmeasured confounders, a bias analysis for unmeasured confounding would be useful. For example, one key confounder not accounted for in this analysis is TB infection (in contrast to TB disease). TB infection is associated with increased risk of multiple non-communicable diseases and mortality risk. The authors could perform a bias analysis due to unmeasured confounding from TB infection or more broadly calculate an e-value to estimate the potential confounding relationships necessary to explain away the main study results.

R: We thank the reviewer for the suggestion. We have now provided e-values for the risk ratios in Table 2 and discussed the implications in the limitations section.

Limitations section:

“We also estimated E-values to assess the robustness of our findings, the e-value for the risk ratio at 10 years was 4.50 for the diagnosed cohort and 3.91 for the treated cohort, considering the strength of confounders in previous studies, the only confounder with effect measures associations greater than 4.00 was age, which was accounted in our study, indicating that the tuberculosis-mortality association found in our study is unlikely to be fully explained by unmeasured confounders.^{6,8,40}”

G. References: appropriate credit to previous work?
References are appropriate.

R1-P10)

H. Clarity and context: lucidity of abstract/summary, appropriateness of abstract, introduction and conclusions

Abstract is appropriate.

The Figures and Tables are difficult to read.

R: We have now revised the tables and figures to improve readability. We reduced the number of variables in table 1, transferring the full table to the supplementary material, and redesign the figure 1.

Reviewer #2 (Remarks to the Author):

Thank you for the possibility to review this manuscript. The authors have described an increased risk of death after diagnosis and treatment completion of tuberculosis. While this is not a new finding, the authors have assessed several new and interesting aspects. Firstly, they have been able to match controls much closer to cases when it comes to socio-economic aspects and also assessed the effect of socio-economic factors itself on the risk of death; thereby showing that the increased risk of death after tuberculosis is not just based on poorer socio-economic circumstances that often underlie tuberculosis. Secondly, they have distinguished between patients diagnosed with TB and patients completing TB treatment, showing that the increased risk of death is not only in those who have not been successfully treated for TB.

The methods have mostly been clearly described and the manuscript is easy to read. The findings are clearly presented and seem straightforward.

R2-P1)It is almost a pity that the authors try to explain their findings by adding in some immunological details (line 410), which seem to be out of place in the context of the manuscript.

R: We have removed the immunological details from the discussion, improving the flow of the discussion.

“Our findings showed that diagnosed and treated TB cases had a higher risk of mortality across a broad range of causes, including respiratory, cardiovascular, endocrine and cancer. Our results complement previous evidence showing that TB is associated with

an increased risk of respiratory mortality, mainly due to direct lung damage caused by TB, increasing the risk of recurrent pneumonia and chronic obstructive pulmonary disease, bronchiectasis and other specific infections, such as aspergillomas.^{3,21} Our findings of sustained increased risk of cardiovascular deaths post-TB diagnosis for more than a decade complements previous epidemiological studies assessing this association on the short-term scale.^{7,22,23}

“Our findings showed that TB is associated with an increased risk of deaths from cancer, including cancer in the digestive organs. This finding is consistent with a growing body of evidence; notably, a prior meta-analysis of 11 studies showed an elevated risk of cancer in TB patients for more than five years after diagnosis.²⁸ Although the mechanisms underlying this association are not fully understood, several hypotheses have been proposed, such as chronic systemic inflammation that can promote carcinogenesis through the promotion of reactive oxygen species and DNA damage,^{29–31} The increased risk of deaths from cancer may also be partially explained by a higher prevalence of shared lifestyle risk factors for cancer and TB, such as smoking and alcohol use.^{8,32–34} Our work extends this body of evidence by providing detailed, year-by-year risk estimates, highlighting the significant long-term outcomes of TB and supporting the need for continued patient care for tuberculosis-related sequelae after successful treatment.”

R2-P2)Line 53-55: WHO end TB strategy: these sentences read a bit weird as we are already halfway through 2025; are there any updated numbers available?

R: We have now updated the numbers of the TB strategy for its end, in 2035:

“These alarming figures highlight the critical importance of the World Health Organisation’s (WHO) End TB Strategy, which aims to reduce TB incidence by 90% and TB mortality by 95% by 2035, relative to 2015 levels.² However, progress has been insufficient: by 2023, the global TB incidence has declined by only approximately 8% and TB mortality has declined by approximately 23%.¹”

R2-P3)Line 105: I would like to see some more details about how the databases were linked. Also, I do not understand what the sensitivity and specificity refer to.

Thank you for pointing this out. We used the CIDACS-RL tool to perform the record linkage. This tool applies a similarity index based on five variables to link datasets in a pairwise manner. The algorithm output includes a probability of that pair being a “true

pair” (ie. they are the same individual). After the linkage is completed, a sample of 2,000 record pairs is randomly drawn, stratified into three categories of linkage scores probabilities: high (>0.95), intermediate ($0.90-0.95$), and low (<0.90). This sample is then manually reviewed, blinded to the assigned probabilities, to assess the quality of the linkage. Sensitivity and specificity are estimated by comparing the threshold of probability that will be considered as true pairs (e.g. $prob>0.95$ of similarity is a true pair) from the linked dataset to the manually reviewed “gold standard”, as usually performed in probabilistic record linkage. So, we compare the pairs considered as true by those above the threshold with the true pairs by the manual review, creating a 2x2 table. We have added more details about the linkage process to the manuscript.

R: More details have now been added to the methods section:

“The linkage between the two databases was done using CIDACS-RL,¹¹ a linkage tool based on a similarity index between registries in different databases. The linkages were based on five variables (name of the individual, name of the mother, date of birth, sex and municipality of residence). The linkage between TB registries and the CadÚnico had a sensitivity (94.6%, showing the proportion of true TB registries classified as TB) and specificity (93.6%, showing the proportion of non-TB registries classified as non-TB) calculated based on false or true links between the two databases. The linkage between mortality registries and the CadÚnico calculated by year had a sensitivity that ranged between 97.8% and 100.0% and a specificity between 96.6% and 99.9% depending on the year of death, details about this linkage can be found in previous article.¹²”

R2-P4) Figure 2: is the figure legend complete?

R: We thank the reviewer for noticing it. We have fixed the legend of the figure:

“Figure 2: Yearly Incidence rate ratios for natural causes deaths (i.e., deaths excluding HIV, TB and external causes) of diagnosed (red) and treated (blue) TB exposed participants compared to unexposed participants. Error bars represent 95% confidence intervals.”

R2-P5) Line 357: why is this cohort so much younger?

R: Thanks you for your query. We agree that the 100 Million Brazilian cohort is younger than most TB cases in Brazil, which mainly occur in adults, due to the characteristics of people applying for social benefits in Brazil. Specifically, our matched cohort (tuberculosis cases matched with household members) is much younger due to the matching in age and sex within the same household, in households living in only one family (with parents of different sex), only the sons/daughters would have the possibility

of matching with a control as the parent wouldn't have a control in the same age range (considering parents of different sex). Adults can only be matched in the cases of households with multiple families living in it.

R2-P6) There are some spelling / text errors (eg. line 408, line 443).

R: We thank the reviewer for pointing this out. We have revised the text, removing typos and improving the grammar

Reviewer #3 (Remarks to the Author):

Manuscript Title: Long-term risk of death after Tuberculosis diagnosis and treatment in Brazil: a nationwide longitudinal study using linked routine data between 2004 and 2018

Journal: Nature Medicine

Dear authors,

I have carefully reviewed the manuscript. The study addresses an important question regarding long-term mortality risk associated with tuberculosis, and the authors leverage a large administrative dataset with detailed socioeconomic information. The overall contribution is potentially valuable for the field. I would like to acknowledge the considerable effort the authors have invested in assembling and analysing this dataset; the exhaustive analyses presented reflect both the importance of the topic and the authors' strong commitment to producing meaningful results.

R: Thank you!

We have grouped some of the points raised by the reviewer, as they deal with correlated topics.

Data Quality: The study is based on secondary data sources. While this inevitably implies the possibility of some missing information or unmeasured variables, the available data appear to be of high quality, as reported by the authors. The dataset and the methods used for extraction and analysis seem reliable and

transparent, and a brief clarification of data completeness could further help readers appreciate the strengths of the information.

R3-P1) Level of Support for the Conclusions: The findings are compelling, but certain clarifications and additional analyses (or discussions of limitations) could make the conclusions more robust. This includes a clearer specification of the study design, a more detailed handling of comorbidities and sociodemographic variables, and an explicit mention of unmeasured factors such as latent TB infection.

R3-P5) Validity: The data and analyses are generally reliable, though some methodological clarifications (study design, handling of comorbidities, and possible misclassification) would enhance robustness. None of these issues preclude publication but should be openly addressed.

R3-P7) Data and Methodology: The dataset is rich and well-documented, though further clarification on completeness, coding of comorbidities, and supplementary analyses would help readers assess reproducibility.

R: We have now improved the methods section related to Tuberculosis and comorbidity data. We also expanded the limitation section about the unmeasured factors

“We used data from the 100 Million Brazilian Cohort linked with nationwide death and TB registries. The 100 Million Brazilian Cohort is a dynamic cohort comprising over 130 million individuals from the Unified Registry for Social Programs (CadÚnico). CadÚnico serves as Brazil's primary tool for identifying and registering low-income families (i.e., families with monthly per capita income less than half minimum wage (approximately 284 USD in 2025) and enrolling them in eligible social welfare programs. Thus, it captures predominantly individuals from the lower socioeconomic strata of the country. We linked the CadÚnico database to TB disease records from Jan 1, 2004, to Dec 31, 2018, registered in the National Notifiable Disease Information System (SINAN) and the Mortality Information System (SIM). The SIM has an estimated completeness of 98% of all deaths occurring in Brazil.¹⁰

In Brazil, TB is a mandatory notification disease, with diagnosis made by rapid molecular test for TB, bacilloscopy, sputum culture, thorax X-ray or clinical case definition.¹¹ All suspected and confirmed cases of TB should be registered with SINAN

via a notification and follow-up form that contains socio-demographic and clinical information about the individuals, including information about being a newly diagnosed TB case or reinfection case and also data about the treatment outcome. This should be filled by a healthcare professional, usually a nurse or a medical doctor.”

“Subgroup analyses were conducted by sex, age at diagnosis of TB (<18, 18-59 and ≥60 years), race (white/mixed/black), type of TB (pulmonary, extrapulmonary and extrapulmonary + pulmonary-both-), diagnosis of HIV, and diagnosis of diabetes mellitus (DM). The covariates related to the diagnosis of HIV and diabetes mellitus were extracted from the SINAN database; i.e., only individuals with a diagnosis of TB have this information. In the subgroup analysis of HIV and DM, the comparison is made between individuals with a diagnosis of TB and a diagnosis of HIV or DM versus the matched unexposed participant.”

“Third, we only had comorbidity data on HIV and DM for the TB group, which were assessed at the time of TB diagnosis via the SINAN system. This information was absent for the unexposed participants in the CadUnico database. Consequently, our subgroup analyses estimate the joint effect of TB plus comorbidities, rather than the effect of TB alone. This prevents us from disentangling the independent effect of TB from the effect of these pre-existing comorbidities.”

R3-P3) Key Results: The results are highly interesting and contribute valuable information to the field. However, despite the detailed analysis, some findings do not appear to be fully supported by the presented evidence, and a stronger alignment between results and conclusions would improve the manuscript.

R: We have now revised the discussions according to specific review comments, which we believe better aligns with the results.

R3-P10) Clarity and Context: The text is generally clear, but the discussion would benefit from additional framing in the context of existing literature on long-term TB outcomes.

R: We have revised the discussion and amended some strengths and limitations which we believe have strengthened the manuscript.

R3-P11) References: The cited literature is relevant, but expanding on prior studies of

mortality in household contacts and latent TB would better situate the work within the broader field.

R: Thank you for the suggestion. There acknowledge the literature on increase risk of mortality among people with latent TB infection (LTBI) [1,2]. Unfortunately, we did not have information on LTBI in our data. In our work, the use of household contacts aimed to define a baseline risk for people with shared socioeconomic characteristics and exposure to TB patients but not exactly a measure of LTBI - i.e., it aimed to estimate residual confounding. We have now clarified that in the methods and in the discussion:

“Our attempt to quantify residual confounding involved conducting an additional analysis to evaluate mortality among household contacts compared to people without TB using the same approach as the main analysis. Although our intent was not to estimate the risk of death in contacts itself, contacts share similar socioeconomic and housing conditions to people with TB, and we do not expect a strong causal link between being a TB contact and an increased risk of death. This analysis revealed a slightly elevated risk of death in this group.”

[1] Wang L, Kuang Y, Zeng Y, Wan Z, Yang S, Li R. Association of systemic inflammation response index with latent tuberculosis infection and all-cause mortality: a cohort study from NHANES 2011-2012. *Front Immunol.* 2025 Feb 19;16:1538132. doi: 10.3389/fimmu.2025.1538132. PMID: 40046059; PMCID: PMC11880221.

[2] Hsu W, Jiang MY. Association of serum vitamin D levels and dietary vitamin D intake with latent tuberculosis infection and long-term mortality: a population-based cohort study. *BMJ Nutr Prev Health.* 2025 Jun 6;8(1):e001213. doi: 10.1136/bmjnp-2025-001213. PMID: 40771511; PMCID: PMC12322562.

Major Comments

R3-P12) 1. Study Design and Terminology

Comment: The manuscript refers to the study as a cohort design. However, the matching procedures on multiple baseline covariates suggest features of a matched cohort or even a nested case–control/case–cohort framework. This distinction influences how results are interpreted.

Suggestion: It may help readers if the authors clarify explicitly how they conceptualize the design (matched cohort vs. nested case–control) and briefly

note its implications for analysis. Alternative descriptions may also be valid, but additional clarity would enhance transparency.

R: The study is a cohort with TB exposed and unexposed participants. The matching was used as the method for confounder adjustment,[1] targeting the average treatment effect on the treated (ATT). We have used exact matching as it is insensitive to modelling assumptions.[1] We use data from individuals registered to receive social programs in Brazil (the baseline data) as the source for the cohort; then linked to the tuberculosis notification system (to define those exposed to TB and those not linked in the source cohort, as not exposed) and the mortality information system. And individuals with/without TB are paired, similar to previous studies.[2-4]

We have changed the wording, use of exposed/unexposed participants instead of case and control, in the text to reduce confusion between cohort and case-control studies.

In the abstract, we changed to

“We conducted a nationwide Brazilian **matched-pair cohort** study using linked data (2004-2018) to quantify long-term mortality (up to 14 years) following:

“From the linked dataset, we built **two matched-pair** cohorts: “

And in the main text we changed “controls” to “unexposed” participants.

[1] Greifer, Noah, and Elizabeth A. Stuart. "Matching methods for confounder adjustment: an addition to the epidemiologist's toolbox." *Epidemiologic reviews* 43.1 (2021): 118-129.

[2] Barda, Noam, et al. "Effectiveness of a third dose of the BNT162b2 mRNA COVID-19 vaccine for preventing severe outcomes in Israel: an observational study." *The Lancet* 398.10316 (2021): 2093-2100.

[3] Cerqueira-Silva, Thiago, et al. "Risk of death following chikungunya virus disease in the 100 Million Brazilian Cohort, 2015–18: a matched cohort study and self-controlled case series." *The Lancet Infectious Diseases* 24.5 (2024): 504-513.

[4] Magen, Ori, et al. "Fourth dose of BNT162b2 mRNA Covid-19 vaccine in a nationwide setting." *New England Journal of Medicine* 386.17 (2022): 1603-1614.

R3-P13) 2. Identification of Prior TB Episodes

Comment: It is unclear whether the dataset allows identification of previous TB

episodes before study entry. If not, there may be a risk of misclassifying recurrent TB as incident, which could influence mortality estimates.

Suggestion: If feasible, specify how prior TB was assessed. If this information is not available, a short discussion of the potential impact and its direction would help readers interpret the findings.

R: Thank you for the opportunity to improve this important point. The Brazilian TB surveillance system already classifies the notified cases as new or recurrent/relapsed. This is done based on the system integration and patient history. As the protocol for diagnosis and treatment, as well as WHO requirement, takes account if the case is new or now to define management, this is well curated in the system. Thus, we , included only those recorded as incident(new) cases in the TB exposed group and excluding those as relapses from the unexposed group. We have now made it clear in the methods section:

“In Brazil, TB is a mandatory notification disease, with diagnosis made by rapid molecular test for TB, sputum culture, thorax X-ray or or clinical case definition.¹⁰ All suspected and confirmed cases of TB should be registered with SINAN via a notification and follow-up form that contains socio-demographic and clinical information about the individuals, including information about being a newly diagnosed TB case or reinfection case and also data about the treatment outcome. This should be filled by a healthcare professional, usually a nurse or a medical doctor.”

R3-P14) 3. Sociodemographic and Household Variables

Comment: The introduction highlights sociodemographic factors, yet the analyses focus mainly on sex, age, and ethnicity. Other socioeconomic and household variables (e.g., housing conditions, household size, registration year) may also provide additional insights.

Suggestion: The authors could consider incorporating these variables more explicitly, perhaps in multivariable models or interaction terms. If this is not possible, a brief acknowledgement of the limitation would guide readers in framing the results.

R: We thank the reviewer for the suggestion. We have now clarified that all of the variables cited in the methods (year of birth, sex, race or ethnicity, city of residence, household location, household water supply type, material of the household, year of registration in the CadÚnico, and household crowding) were used in the confounder

adjustment through exact matching of TB exposed participants and unexposed participants, which provides balance without relying on modelling assumptions. These socioeconomic variables provide value in balancing the groups to more similar socioeconomic conditions, but only when evaluated as a whole, which precludes subgroup analysis. Furthermore, variables such as age, sex, and ethnicity have relatively standardized definitions, facilitating comparison with other research. Conversely, the specific socioeconomic indicators used in our study (e.g., 'material of the household,' 'water supply type') suffer from significant measurement heterogeneity across the literature. This lack of standardized codification means that any subgroup analysis we might perform on them would not be reliably comparable to findings from other studies.

R3-P9) Suggested Improvements: Additional clarifications, some targeted sensitivity analyses, and a more explicit discussion of limitations would be valuable to reinforce the conclusions within the current scope of the manuscript.

R3-P15) 4. Comorbidities and Temporal Dynamics

Comment: It is not clear whether comorbidities (such as diabetes or HIV) were measured dynamically during follow-up or only at baseline. This matters because comorbidities may arise after TB diagnosis and affect mortality risk differently from pre-existing conditions.

Suggestion: If longitudinal information is available, distinguishing pre-existing from incident comorbidities could be informative. If not, a short clarification in the methods and a note in the discussion would be helpful.

R: We thank the reviewer for pointing this out. As our baseline data is from an administrative database related to social programs, there is no information regarding comorbidities. We can only assess the comorbidities in the individuals notified with Tuberculosis through the SINAN notification system. We have included this clarification in the methods and limitations sections:

“Subgroup analyses were conducted by sex, age at diagnosis of TB (<18, 18-59 and ≥60 years), race (white/mixed/black), type of TB (pulmonary, extrapulmonary and extrapulmonary + pulmonary-both-), diagnosis of HIV, and diagnosis of diabetes mellitus (DM). The covariates related to the diagnosis of HIV and diabetes mellitus were extracted from the SINAN database; i.e., only individuals with a diagnosis of TB have this information. In the subgroup analysis of HIV and DM, the comparison is made

between individuals with a diagnosis of TB and a diagnosis of HIV or DM versus the matched unexposed participant.”

“Third, we only had comorbidity data on HIV and DM for the TB group, which were assessed at the time of TB diagnosis via the SINAN system. This information was absent for the unexposed participants in the CadUnico database. Consequently, our subgroup analyses estimate the joint effect of TB plus comorbidities, rather than the effect of TB alone. This prevents us from disentangling the independent effect of TB from the effect of these pre-existing comorbidities.”

R3-P2) Significance of the Results: The results are relevant, interesting, and necessary to advance understanding of the long-term impact of tuberculosis. By refining some analytic choices and clarifying design considerations, the study has strong potential to move the field forward.

R3-P8) Analytical Approach: The use of stratified analyses provides granularity, but it introduces potential challenges related to multiplicity. Considering alternative multivariable approaches could complement the findings.

R3-P16) 5. Multiplicity and Stratification

Comment: The number of stratified analyses (by sex, age, ICD chapters, ICD blocks) provides detail but increases the risk of multiplicity and small sample sizes in some strata.

Suggestion: An alternative could be to employ multivariable competing-risk models (e.g., Fine–Gray regression) that handle several covariates simultaneously. This might reduce the multiplicity burden while maintaining interpretability. If the authors prefer the current approach, a brief discussion of its trade-offs would still be valuable.

R: We thank the reviewer for the suggestion. However, the stratification was used to assess the different effects of tuberculosis within each group, not to adjust for those characteristics. We have used the matching as the confounder adjustment strategy, as our exposure is time-varying characteristic (TB diagnosis or TB treatment). Using conventional multivariable approaches with time-varying exposures in the Fine and Gray model changes the interpretation of the model, not allowing to interpret the coefficients as changes in the cumulative incidence function.[1] Also, the estimation of the TB effect by subgroup using the fine and gray model would lead to similar

multiplicity issues. In addition, the fine and gray model is a semiparametric model that uses the same estimator (Aalen-Johansen) but it requires the proportional assumption of the subdistribution hazards.[2] While the use of the Aalen-Johansen estimator to estimate cumulative incidence functions/risks in the matched analysis is a non-parametric estimator with no assumption of proportionality.[2]

[1]Austin, Peter C., Aurélien Latouche, and Jason P. Fine. "A review of the use of time-varying covariates in the Fine- Gray subdistribution hazard competing risk regression model." *Statistics in medicine* 39.2 (2020): 103-113.

[2] Geskus, Ronald B. "Competing risks: concepts, methods, and software." *Annual Review of Statistics and Its Application* 11 (2015).

R3-P17) 6. Contacts as a Separate Group

Comment: Household contacts are analysed separately, but direct comparisons between TB cases and contacts are not shown. This may limit the ability to distinguish effects of TB disease from those of the shared environment.

Suggestion: If possible, including direct comparisons of cases and contacts (using matching or propensity score methods) would provide further insights. Alternatively, a short explanation of why this was not pursued would guide the reader.

R: We have included the direct comparison between diagnosed cases and household contacts, in this analysis we limited the matching characteristics to only sex and age. This cohort (tuberculosis cases matched with household members) is much younger due to the matching in age and sex within the same household, in households living in only one family (with parents of different sex), only the sons/daughters would have the possibility of matching with a control as the parent wouldn't have a control in the same age range.

In the results section:

"In the second analysis, a total of 12,948 (6.2%) TB cases were matched to household contacts. The median age of the pairs was 14 years (IQR 11 to 18), and 8,980 pairs (69.4%) were male. The direct comparison between TB cases and household contacts for natural deaths yielded results to those of the main analysis, RR of 11.15 (6.49 to 23.10), 3.97 (2.94 to 5.70) and 2.52 (1.83 to 3.66), at 1, 5 and 10 years, respectively. For external causes of death, the RR at 1, 5 and 10 years were 2.29 (1.48 to 4.09), 1.66 (1.29 to 2.12) and 1.40 (1.06 to 1.84), respectively. (Supplementary Table 16)"

This is also discussed in...:

“However, similarities in the RR between the treated and diagnosed cohorts reinforce the plausibility of these increased risks being due to a societal rather than biological phenomenon. In addition, our direct comparison of TB cases to household contacts of the same sex and similar age also showed elevated risk of deaths from external causes in the TB group. This finding indicates that, even under similar socioeconomic conditions, TB patients experience an increased risk of death from external causes; however while some of this association can be related to residual confounding, it is unlikely to account for all of it. In this context, increased awareness of TB stigma and interventions to address it, such as TB support groups, training healthcare workers to provide nonjudgmental care, and community-wide educational campaigns to dispel myths and misinformation about TB can improve the lives of TB survivors.³⁹”

R3-P6) Significance: The conclusions have the potential to influence understanding of long-term TB outcomes. Greater clarity regarding confounding factors and the role of latent infection would strengthen the study’s impact.

R3-P18) 7. Latent TB Infection, Reactivation, and Reinfection

Comment: The study does not directly measure LTBI or distinguish reactivation from reinfection, yet these factors are central for interpreting mortality in contacts.

Suggestion: It could strengthen the discussion to acknowledge this explicitly and, if feasible, include a sensitivity or bias analysis using plausible LTBI prevalence estimates. If this is not possible, even a qualitative discussion would help contextualize the findings.

R: The exposed participants are only those classified as a new TB case in the notification form, as described in the R3-P13).

“In Brazil, TB is a mandatory notification disease, with diagnosis made by rapid molecular test for TB, bacilloscopy, sputum culture, thorax X-ray or or clinical case definition.¹⁰ All suspected and confirmed cases of TB should be registered with SINAN via a notification and follow-up form that contains socio-demographic and clinical information about the individuals, including information about being a newly diagnosed

TB case or reinfection case and also data about the treatment outcome. This should be filled by a healthcare professional, usually a nurse or a medical doctor.”

However, we have no information about LTBI, the implications of these limitations have been included in the discussion section:

“Sixth, we lack information about latent tuberculosis infection (LTBI), which can result in some individuals with LTBI being classified as TB-free unexposed participants, biasing our estimates downwards as previous studies have shown that LTBI slightly increases the risk of death compared to healthy individuals.⁴⁵”

Minor Comments

R3-P19) The supplementary tables are extensive. The authors may consider consolidating some material or highlighting the most essential comparisons in the main text to improve readability.

R: We have included more information in the results section, directing to specific findings.

“The cause-specific analysis showed a similar pattern to that of the analysis of natural death. In the diagnosed cohort, the highest RRs were observed within 30 days of the diagnosis, which decreased over time. (Supplementary Tables 4, 7 and Supplementary Figure 2) The cumulative incidence of cardiovascular, cancer and respiratory deaths were similar for the TB exposed group, while the cardiovascular deaths were more incident in the unexposed participants. (Supplementary Figure 2)

In the treated cohort, the highest RR for cancer and respiratory deaths were seen within 90 days of treatment completion. (Supplementary Tables 5, 8 and Supplementary Figure 4) The cumulative incidence of cardiovascular deaths were slightly higher than cancer and respiratory deaths. (Supplementary Figure 4)”

“The treated cohort showed lower RRs for all cause-specific deaths, especially for cancer and respiratory diseases, for example the one year RR for respiratory deaths in the diagnosed cohort was 12.72 (10.21 to 16.93), and 6.15 (4.71 to 8.78) in the treated cohort.

Notably, within the specific cancer types, we found an increased risk for cancer of the digestive organs in both cohorts; the RR at 14 years were 1.97 (1.55 to 2.57) in the diagnosed cohort and 1.98 (1.40 to 2.89) in the treated cohort. (Supplementary Tables 7 and 8).

One exception that presented similar values across both cohorts was death from external causes. In the diagnosed cohort, the RR values ranged from 1.72 to 2.47, and 1.74 to 2.10 in the treated cohort. (Figure 3 and Supplementary Tables 4 and 5). In the analysis by ICD-10th blocks of external causes, deaths from assault (homicides) presented higher RR than deaths from accidents in both cohorts. (Supplementary Table 7 and 8)”

R3-P20) Consistent terminology for “diagnosed cohort,” “treated cohort,” and “household contacts” would aid clarity. Including E-values or another quantitative measure of robustness could further support interpretation.

R: We have now reviewed the text, to use the term consistently. We have also included E-values for Table 2 and included in the discussion the interpretation related to the e-values

“We also estimated E-values to assess the robustness of our findings, the e-value for the risk ratio at 10 years was 4.50 for the diagnosed cohort and 3.91 for the treated cohort, considering the strength of confounders in previous studies, the only confounder with effect measures associations greater than 4.00 was age, which was accounted in our study, indicating that the tuberculosis-mortality association found in our study is unlikely to be fully explained by unmeasured confounders.^{6,8,40”}

R3-P21) Finally, a note on whether household clustering was addressed (e.g., through multilevel modelling) would reassure readers about potential shared vulnerability.

R: Our model doesn't use mixed effects, as we use the non-parametric Aalen johansen estimator. The shared vulnerability was accounted for by matching persons by household characteristics (water supply, overcrowded, material, etc) - this means that clustering at household level was accounted by design in this process. We have also used bootstrap for the estimation of the confidence intervals, providing valid CI in this scenario.

Reviewer 4 comments

This manuscript presents significant findings; although retrospective in nature, the information provided is valuable in understanding mortality related to tuberculosis diagnosis, as well as outcomes following completion of treatment and bacteriological cure.

The introduction is clear and concise, and the goal of the study is established.

R4-P1) Methods: Data from three separate registries in Brazil were cross-referenced, and two cohorts were extracted from these combined databases: Diagnosed Cohort and treated cohort

CadUnico: registry for low income families, how is this defined?

R: We have now clarified eligibility criteria for being registered with CadUnico in the methods section

“The 100 Million Brazilian Cohort is a dynamic cohort comprising over 130 million individuals from the Unified Registry for Social Programs (CadÚnico). CadÚnico serves as Brazil's primary tool for identifying and registering low-income families (i.e., families with monthly per capita income less than half minimum wage (approximately 284 USD in 2025) and enrolling them in eligible social welfare programs.”

SINAN: TB registry

It is important to include how TB diagnosis is made.

R: We have now clarified that information in the methods section

“In Brazil, TB is a mandatory notification disease, with diagnosis made by rapid molecular test for TB, bacilloscopy, thorax X-ray or or clinical case definition.¹⁰ All suspected and confirmed cases of TB should be registered with SINAN via a notification and follow-up form that contains socio-demographic and clinical information about the individuals, including information about being a newly diagnosed TB case or reinfection case and also data about the treatment outcome. This should be filled by a healthcare professional, usually a nurse or a medical doctor.”

R4-P2) All included cases were microbiologically confirmed or suspected cases were also included?

R: Thank you for your question. Not all cases were microbiologically confirmed, but 69% of our diagnosis cohort was positive for at least one laboratory diagnosis test, culture (16%) or histopathology (10%) or bacilloscopic (56%) or molecular test (10%). Limiting the analysis to cases with culture would induce substantial selection bias to individuals, not only to high-risk groups, but to individuals living in large cities that have better healthcare structures. As the guidelines of the Brazilian Thoracic Association in 2009, recommended culture only in specific cases, such as negative bacilloscopic, co-infection with HIV, suspicion of resistance, etc. Last, it is needed to consider the Brazilian scenario (high burden TB, with limited infrastructure for laboratory diagnosis-culture)).

We now acknowledge this in the limitation section:

“Fourth, only 15% of the TB cases had sputum culture positive, which is considered the gold standard for the diagnosis of TB. This proportion reflects the Brazilian guidelines, which prioritises sputum culture for suspected TB cases that present a negative bacilloscopy, and specific cases such as suspected cases of resistance, retreatment cases, in prison populations.⁴² In addition, due to the required infrastructure and time for diagnosis, the use of sputum culture as primary diagnosis for TB remains limited in LMICs.^{43,44} The inclusion of non confirmed TB patients in the TB group likely underestimate the risk in this group.”

For the treated cohort only patients with positive outcome were included?

R: For the treated cohort, only patients classified as “treatment success” and with a compatible date of conclusion (between 138 days and 2 years) were included. We have clarified it in the methods:

“We defined treatment completion as being classified as “treatment success” in the SINAN. Patients with treatment completion are those who have two negative smear tests according to national guidelines or who do not have evidence of treatment failure based on clinical or radiological criteria.10”

“For the treated cohort, we only included TB cases that were classified as “treatment success” in SINAN-TB. To remove records with inconsistent treatment length, we also excluded individuals whose treatment completion date was missing, less than 138 days after the notification date or more than 2 years after the notification date. We used this cut-off considering the treatment duration of 6 - 12 months, as recommended by the Ministry of Health in Brazil, depending on the type of Tuberculosis, the maximum of two years was chosen to allow for delays in the start of treatment after diagnosis.¹⁰”

For the treated cohort, there is no minimum number of doses??

R: Unfortunately, there is no information about the number of doses each patient took before being considered cured/treatment success. However, by definition, the minimum treatment of 6-month doses is defined to consider treatment success.

R4-P3) Patients with only one TB event were included? Relapse cases were not included? This is important information as relapse has an impact on mortality.

R: We have included only newly diagnosed cases. We have now clarified it in the methods section:

“In Brazil, TB is a mandatory notification disease, with diagnosis made by rapid molecular test for TB, bacilloscopy, sputum culture or thorax x-ray.¹⁰ All suspected and confirmed cases of TB should be registered with SINAN via a notification and follow-up form that contains socio-demographic and clinical information about the individuals, including information about being a newly diagnosed TB case or reinfection case and also data about the treatment outcome. This should be filled by a healthcare professional, usually a nurse or a medical doctor.”

R4-P4) All treated cases received the same regimen?

R: Thank you for this query. Yes, almost all TB cases in Brazil are treated with a standard 2 months of RHZE (R – Rifampicina; H – isoniazida; Z – Pirazinamina; E – Etambutol) followed by 4 months of treatment maintenance with RH (Rifampicina and isoniazida) during the study period. Deviations from there include in rare cases on intolerance to one drug and those changes recorded on SINAN registration system, as well as a small proportion of treatments ending in drug-resistance which were excluded from the treated cohort. However, the end of treatment registration stated as “treatment success” is not influenced by changes in treatment but followed by guidelines of having

completed negative sputum smear followed by 4 months of treatment maintenance with RH.

The inclusion criteria and matched controls are explicitly described.

Subgroup analysis was conducted by sex, age at diagnosis, race, type of TB Pulmonary, extrapulmonary and pulmonary + extrapulmonary. It is essential to specify the number of CNS-TB patients included, as this is a severe form of tuberculosis and associated with increased mortality and morbidity.

R: We agree meningoencephalic TB is a severe form of TB, including high short-term mortality. However, they represent a small proportion of TB cases (as now showed in supplementary Table 2), and so unable to estimate mortality post diagnosis or treatment, especially those not related to TB itself. CNS cases represent 1,527 (5.4%) out of the 28,312 extrapulmonary TB in the diagnosed cohort and 442 (2.8%) in the treated cohort. Most of the extrapulmonary TB were located in the Pleura (~42%) and lymph nodes (~22%).

R4-P5) Were all included cases drug sensitive? No drug-resistant cases were included?. This distinction is relevant because drug-resistant tuberculosis typically presents more treatment challenges and is associated with higher mortality rates. Brazil also reports a significant number of drug-resistant TB cases.

R: We have also included cases of drug resistant (DR-TB) but not multidrug-resistant (MDR-TB) as the latter is registered in a different system. However, in our analysis of treated cases, this would only include sensitive or DR-TB who were considered cured. Importantly, drug resistant cases are rare in Brazil.

We have included this in the limitation:

“Fifth, we lack detailed information (type of resistance) about the TB cases that were identified as drug-resistant (n = 945; 0.5%) after completing the treatment scheme, considering the very low numbers it is unlikely to significant bias the analysis of diagnosed cases, and it has no effect in the analysis of treated cases, as these TB cases were not included”

R4-P6) Household contacts cohort: How many individuals were screened for latent tuberculosis, and how many received preventive treatment? Also important for mortality and morbidity

R: Unfortunately, we don't have information on TB screening of all household contacts. However, a previous study from our group using the same cohort of linked data detected around 9,000 active TB cases among over 420,000 household contacts followed up for a median of 4.4 years (incidence of 427.8 per 100,000 pyr) between 2004 and 2018 [1]. This reflects both active and passive case finding. In addition, it is important to mention that although investigation of active and latent-TB among household contacts is suggested, treatment for ILTB was not protocolized in the study period, and was indicated them only in certain cases, e.g., when the case was a MRD-TB. In addition, contact tracing is quite low. We have included this in the limitation section:

“Sixth, we lack information about latent tuberculosis infection (LTBI), which can result in some individuals with LTBI being classified as TB-free unexposed participants, biasing our estimates downwards as previous studies have shown that LTBI slightly increases the risk of death compared to healthy individuals.”⁴⁵

[1] Pinto, Priscila FPS, et al. "Incidence and risk factors of tuberculosis among 420 854 household contacts of patients with tuberculosis in the 100 Million Brazilian Cohort (2004–18): a cohort study." *The Lancet Infectious Diseases* 24.1 (2024): 46-56.

R4-P7) The diagnosed cohort was followed from diagnosis until death or December 31, 2018; the treated cohort was followed from treatment completion until death or December 31, 2018. Was there no minimum follow-up period?

R: We did not restrict individuals with a minimum of follow-up time, as we have used time to event analysis, making the contribution of each individual being proportional to their time of follow-up.

R4-P8) Does the study has ethical approval?

R: We thank the reviewer for noticing the lack of ethics paragraph. The study has ethical approval, we have now included the paragraph in the manuscript.

“The study was approved by the ethics committees of the Instituto Gonçalo Muniz–Oswaldo Cruz Foundation (1.612.302), Salvador, Brazil.”

Results:

R4-P9) The median follow-up duration differs between the diagnosed and treated cohorts and the control groups. Specifically, the controls have a shorter follow-up period compared to the diagnosed and/or treated cohorts. This discrepancy may be attributable to higher mortality rates among the control group?

R: The median follow-up for the treated and diagnosed exposed participants are shorter. We have now rephrased the text to avoid confusion.

The reason is the very high mortality within one year of the tuberculosis group, in the diagnosed group there are 4636 deaths versus 673 in the control group in that cohort.

“The median follow-up period in the diagnosed cohort was 4.5 years (interquartile range – IQR 1.9 to 7.7) for the unexposed group and 3.9 years (IQR 1.5 to 7.2) for the diagnosed group. For the treated cohort, the median follow-up was 4.0 years (IQR 1.8 to 7.2) for the unexposed group and 3.8 years (IQR 1.7 to 7.0) for the treated group. The median time to treatment completion was 6.6 months (IQR, 6.2 to 7.7).”

R4-P10) Regarding cause-specific mortality, it is important to highlight which causes are more prevalent within each cohort.

R: We thank the reviewer for the suggestion, we have included cumulative incidence functions for all main causes evaluated, and highlighted in the results in the text.

“The cause-specific analysis showed a similar pattern to that of the analysis of natural death. In the diagnosed cohort, the highest RRs were observed within 30 days of the diagnosis, which decreased over time. (Supplementary Tables 4, 7 and Supplementary Figure 2) The cumulative incidence of cardiovascular, cancer and respiratory deaths were similar in the TB exposed group, while the cardiovascular deaths had the highest incidence in the unexposed participants. (Supplementary Figure 2) In the treated cohort, the highest RR for cancer and respiratory deaths were seen within 90 days of treatment completion. (Supplementary Tables 5, 8 and Supplementary Figure 4) The cumulative incidence of cardiovascular deaths were slightly higher than cancer and respiratory deaths. (Supplementary Figure 4)”

Cause of De

Supplementary Figure 2. Cumulative incidence curve of cause-specific mortality in the diagnosed tuberculosis cases and the unexposed control group. a) HIV and TB deaths,

b) Cancer, endocrine, respiratory and cardiovascular deaths, c) Natural deaths (i.e., deaths excluding HIV/TB and external causes)

A

B

Tuberculosis: — Exposed - - Unexposed

Cause of De

Supplementary Figure 4. Cumulative incidence curve of cause-specific mortality in the treated tuberculosis cases and the unexposed control group. a) HIV and TB deaths, b) Cancer, endocrine, respiratory and cardiovascular deaths, c) Natural deaths (i.e., deaths excluding HIV/TB and external causes)

R4-P11) “Participants with diabetes mellitus had a similar RR to those with HIV in the diagnosed cohort and a higher RR in the treated cohort. Notably, the absolute excess mortality (risk difference) in TB patients with DM was substantially larger, exceeding double the RD observed in TB patients with HIV”.

This paragraph refers to comparisons made with patients diagnosed and treated for TB, rather than with control groups, as the control groups do not specify the presence of DM or HIV.

R: This paragraph is related to the comparison between diagnosed TB exposed and unexposed TB participants (and treated TB exposed and unexposed participants), as we don't have information about comorbidities in the control group, this comparison reflects the joint effect of TB+DM or TB+HIV. We have now made it clear in the methods section and in the limitation is stated in the discussion section

“Subgroup analyses were conducted by sex, age at diagnosis of TB (<18, 18-59 and ≥60 years), race (white/mixed/black), type of TB (pulmonary, extrapulmonary and extrapulmonary + pulmonary-both-), diagnosis of HIV, and diagnosis of diabetes mellitus (DM). The covariates related to the diagnosis of HIV and diabetes mellitus were extracted from the SINAN database; i.e., only individuals with a diagnosis of TB have this information. In the subgroup analysis of HIV and DM, the comparison is made between individuals with a diagnosis of TB and a diagnosis of HIV or DM versus the matched unexposed participant.”

“Third, we only had comorbidity data on HIV and DM for the TB group, which were assessed at the time of TB diagnosis via the SINAN system. This information was absent for the unexposed participants in the CadUnico database. Consequently, our subgroup analyses estimate the joint effect of TB plus comorbidities, rather than the effect of TB alone. This prevents us from disentangling the independent effect of TB from the effect of these pre-existing comorbidities.”

R4-P12) The results indicate an association between TB diagnosis and TB-treated cases with cancer, especially gastrointestinal cancer.

Patients are matched with controls based on age, sex, and socioeconomic status, rather than comorbidities. It is important to include (if this information is available) that no patient had a prior diagnosis of cancer.

R: Unfortunately, we don't have information about comorbidities for all individuals recorded in the CadUnico, the information about comorbidities was extracted from the Tuberculosis notification system. We have changed the method section to make it clear. This limitation is also stated in the discussion section

“Subgroup analyses were conducted by sex, age at diagnosis of TB (<18, 18-59 and ≥60 years), race (white/mixed/black), type of TB (pulmonary, extrapulmonary and extrapulmonary + pulmonary-both-), diagnosis of HIV, and diagnosis of diabetes mellitus (DM). The covariates related to the diagnosis of HIV and diabetes mellitus were extracted from the SINAN database; i.e., only individuals with a diagnosis of TB have this information. In the subgroup analysis of HIV and DM, the comparison is made between individuals with a diagnosis of TB and a diagnosis of HIV or DM versus the matched unexposed participant.”

“Second, our results may be subject to reverse causality, particularly for cancer and endocrine-related mortality. Given that active malignancies and diabetes are known risk factors for TB and that we were unable to adjust for pre-exposure comorbidity, it is possible that an undiagnosed underlying condition precipitated the onset of active TB in some individuals.^{21,26}”

“Third, we only had comorbidity data on HIV and DM for the TB group, which were assessed at the time of TB diagnosis via the SINAN system. This information was absent for the unexposed participants in the CadUnico database. Consequently, our subgroup analyses estimate the joint effect of TB plus comorbidities, rather than the effect of TB alone. This prevents us from disentangling the independent effect of TB from the effect of these pre-existing comorbidities.”

DISCUSSION:

R4-P13) Line 389: “Relying solely on time after diagnosis can group individuals who abandoned treatment, experienced treatment failure, who are classified as drug-resistant TB, or were lost to follow-up with those who completed treatment, likely biasing the results”. This sentence seems incomplete

R: We thank the reviewer for pointing this out. We have rephrased the sentence

“Relying solely on time after diagnosis for classification of the individuals can group those who actually completed the treatment with those who abandoned treatment, experienced treatment failure, who are classified as drug-resistant TB, or were lost to follow-up, likely overestimating the risk of death after treatment completion”

R4-P14) This section details the number of drug-resistant cases included. It is necessary to incorporate this information in the results section and clarify the type of resistance, as outcomes may vary accordingly.

R: Unfortunately, we don't have this type of information. We have expanded the limitation section to describe it:

“Fifth, we lack detailed information (type of resistance) about the TB cases that were identified as drug-resistant (n = 945; 0.5%) after completing the treatment scheme, considering the very low numbers it is unlikely to significant bias the analysis of diagnosed cases, and it has no effect in the analysis of treated cases, as these TB cases were not included.”

R4-P15) Limitations of the study are clearly stated. Another limitation of the study is the absence of control group for DM and HIV cases

R: We have now clarified the comparison group for DM/HIV analysis, both in the methods section and also in the limitation section:

“Subgroup analyses were conducted by sex, age at diagnosis of TB (<18, 18-59 and ≥60 years), race (white/mixed/black), type of TB (pulmonary, extrapulmonary and extrapulmonary + pulmonary), diagnosis of HIV, and diagnosis of diabetes mellitus (DM). The covariates related to the diagnosis of HIV and diabetes mellitus were extracted from the SINAN database; i.e., only individuals with a diagnosis of TB have this information. In the subgroup analysis of HIV and DM, the comparison is made between individuals with a diagnosis of TB and a diagnosis of HIV or DM versus the matched unexposed participant.”

“Third, we only had comorbidity data on HIV and DM for the TB group, which were assessed at the time of TB diagnosis via the SINAN system. This information was absent for the unexposed participants in the CadUnico database. Consequently, our subgroup analyses estimate the joint effect of TB plus comorbidities, rather than the effect of TB alone. This prevents us from disentangling the independent effect of TB from the effect of these pre-existing comorbidities”

R4-P16) Conclusions not so clear, should be rephrased

R: We have rewritten the conclusion, making it more coherent:

“One fundamental limitation about the management of tuberculosis worldwide is the sole focus on diagnosing and curing active disease, overlooking long-term health consequences. For decades, WHO guidelines have appropriately emphasised the diagnosis and bacteriological cure of active disease, considering this a complete return to health, without considering possible post-TB complications. Our findings strongly support the need for long-term clinical follow-up into routine TB care. Integrating post-TB assessments, such as lung function testing, cardiovascular risk screening, and cancer surveillance, into national guidelines for post-TB management is essential. Such measures will enhance clinician awareness of post-TB complications, ensure timely management, and direct resources towards truly comprehensive, patient-centred care.”

R4-P17) Figures and tables:

Figure 2: “Yearly incidence rate ratios from natural causes “ it is missing one word maybe death

R: We thank the reviewer for noticing it. We have fixed the legend of the figure:

“Figure 2: Yearly Incidence rate ratios for natural causes deaths (i.e., deaths excluding HIV, TB and external causes) of diagnosed (red) and treated (blue) TB exposed participants compared to unexposed participants. Error bars represent 95% confidence intervals.”

Remarks to the Author:

Dear Authors,

I have carefully reviewed the revised version of the manuscript. Taking into account both my previous comments and those raised by the other reviewers, I believe that this new version is substantially strengthened. The study is now more clearly framed, its limitations are more explicitly acknowledged, and the conclusions are appropriately supported by the presented results.

I appreciate the authors' thorough and thoughtful responses to my questions and observations. In my view, the manuscript in its current form meets the methodological standards and scientific quality required for publication.

That said, I would like to offer two final, minor suggestions that could further strengthen the interpretation of the results. I consider these optional and leave their inclusion or modification entirely at the authors' discretion.

First, regarding latent tuberculosis infection (LTBI) and residual confounding, I suggest considering the inclusion of the following statement in the Limitations section, to clarify the interpretation of the estimated effects:

“Thus, our results reflect excess mortality following a TB disease episode, not the isolated biological effect of TB, as LTBI and related pre-existing social and health vulnerabilities are unmeasured.”

Second, to further guide the interpretation of the subgroup and cause-specific analyses and to address concerns related to multiplicity, I suggest adding a brief clarification in the Limitations section, such as:

“Accordingly, subgroup and cause-specific estimates should be interpreted as descriptive assessments of heterogeneity and patterning rather than as confirmatory evidence for differential effects.”

R: We thank the reviewer for the positive feedback. We have include the following sentences in the limitation section:

This also represents that our results reflect excess mortality following a TB disease episode, not the isolated biological effect of TB

The subgroup and cause-specific analyses weren't adjusted for multiplicity adjustment, as specified in the analysis plan, and should not be used in place of hypothesis testing